https://doi.org/10.1038/s42003-023-05534-2　　**OPEN**
# Systematic investigation of recipient cell genetic requirements reveals important surface receptors for conjugative transfer of IncI2 plasmids

Nancy Allard[1], Arianne Collette[1], Josianne Paquette[1], Sébastien Rodrigue [1✉] & Jean-Philippe Côté [1✉]

Bacterial conjugation is a major horizontal gene transfer mechanism. While the functions encoded by many conjugative plasmids have been intensively studied, the contribution of recipient chromosome-encoded genes remains largely unknown. Here, we analyzed the genetic requirement of recipient cells for conjugation of IncI2 plasmid TP114, which was recently shown to transfer at high rates in the gut microbiota. We performed transfer assays with ~4,000 single-gene deletion mutants of *Escherichia coli*. When conjugation occurs on a solid medium, we observed that recipient genes impairing transfer rates were not associated with a specific cellular function. Conversely, transfer assays performed in broth were largely dependent on the lipopolysaccharide biosynthesis pathway. We further identified specific structures in lipopolysaccharides used as recipient cell surface receptors by PilV adhesins associated with the type IVb accessory pilus of TP114. Our strategy is applicable to study other mobile genetic elements and understand important host cell factors for their dissemination.

[1] Département de biologie, Faculté des sciences, Université de Sherbrooke, Sherbrooke, QC J1K 2R1, Canada. ✉email: Sebastien.Rodrigue@USherbrooke.ca; Jean-Philippe.Cote2@USherbrooke.ca

Conjugation is a promiscuous mechanism of horizontal gene transfer in bacteria[1], enabling microbes to respond and adapt to their environment through the acquisition of large amounts of genetic material[2]. Conjugative transfer occurs when a donor bacterium assembles a Type IV Secretion System (T4SS) and transfers DNA into a recipient bacterium in a contact-dependent manner[3–6]. Despite the large amount of information accumulated on the genetic function and molecular architecture of conjugative systems[7–9], much remains to be understood about how chromosome-encoded genes influence conjugation. For instance, the genetic requirements of recipient cells for conjugative transfer have not been explored extensively and could vary between different mobile genetic elements. Receptor molecules recognized by the conjugative pilus or other accessory pili involved in mating pair formation (MPF) and stabilization are likely to be required at the surface of a recipient bacterium, but other genes involved in defense mechanisms[10,11], DNA replication[4,6], and gene expression[12,13] could also affect the uptake and maintenance of mobile genetic elements. A powerful but still underexploited approach[1,14] to systematically evaluate the impact of bacterial recipient genes for conjugative plasmid maintenance and replication consists of using ordered single-gene mutant libraries such as the Keio collection comprised of ~4000 mutants[15].

TP114 was isolated from an *Escherichia coli* strain in Scotland[16] and is part of the IncI2 plasmid incompatibility group[17], which has received less attention than other model plasmids such as F[18], RP4[19–21], or R388[22]. Conjugative plasmid TP114 was recently shown to encode a highly proficient DNA transfer machinery reaching transfer rates close to 100% with target *E. coli* bacterial populations present in the gut microbiota[23]. A probiotic bacterium harboring a modified version of this plasmid even achieved full clearance of *Citrobacter rodentium* by high-efficiency delivery of a CRISPR-*cas9* system in a mouse infection model[24].

In addition to a T4SS required for conjugative transfer that is part of the MPF group T, TP114 also encodes a Type IVb Pilus (T4Pb). This T4Pb stabilizes the mating pair rather than being directly involved in DNA transfer (Fig. 1a, b) and is also found in other members of the I-complex plasmid group (including IncB/O, IncI1α, IncI1γ, IncI2, IncK, and IncZ). The T4Pb forms a helical fiber distinct from the T4SS that is not ancestrally related to conjugative pili, but rather phylogenetically and functionally associated with the type II secretion systems and archaeal flagella[25]. The pilus fiber is mainly composed of major pilin subunits (PilS) with additional minor pilins (PilV) acting as adhesins. The T4Pb of TP114 plays a major role in conjugation in broth as well as in the gut microbiota but is dispensable on a solid support[26].

A remarkable feature of the T4Pb encoded by I-complex conjugative plasmids is the presence of a multiple DNA inversion system, called shufflon, spanning from the distal section of the *pilV* gene to the *rci* gene, which encodes a site-specific recombinase also known as a shufflase[27]. The *pilV* gene is followed by multiple DNA cassettes, flanked by *rci*-recombination motifs, that can be swapped independently or in groups to change the 3'-end of *pilV*[28]. Depending on T4Pb, between 2 to 8 different PilV variants can be formed, of which one is thought to be displayed at the tip of the pilus[26,29]. PilV variants appear to recognize specific structures on the surface of recipient cells allowing conjugation in unstable environments where mobility, flow forces, and external factors may prevent bacterial interactions[26]. In IncI1 conjugative plasmid R64, specific disaccharides found in the lipopolysaccharide (LPS) have been identified as the receptors of some adhesins[30,31]. Although LPS make up ~75% of the total membrane surface, other structures like enterobacterial common

antigen, capsular polysaccharides, porins, and other proteins[32] could potentially act as receptors for adhesins encoded by conjugative elements. For example, F plasmid family members recognize outer membrane proteins at the surface of the recipient cell (OmpA, OmpF, OmpK36, OmpW)[33,34] and their transfer efficiencies can be affected by mutation in the LPS biosynthesis pathway[35].

In this work, we used a high-throughput conjugation assay to systematically investigate the genetic determinants encoded by recipient bacteria for efficient TP114-mediated conjugation. Our genome-wide screen was performed on LB solid medium and in broth using the Keio collection[15] and small RNA or small protein deletion mutants[36]. Our results indicated that genes involved in LPS biosynthesis affect the ability of *E. coli* to receive conjugative plasmid TP114 when matings are performed in broth. We also confirmed the critical role of the T4Pb encoded by TP114 to stabilize the interaction between the donor and recipient bacteria under these conditions, and located receptors recognized by PilV adhesin variants mostly in the core section of the LPS.

## Results

**High-throughput screening of genetic requirements in recipient bacteria.** We sought to identify the recipient cell genes important for the transfer of conjugative plasmid TP114 using a high-throughput conjugation assay in a genome-wide screen of ~4000 *E. coli* BW25113 single-gene deletion mutants composing the Keio collection[15] along with 141 small RNA or small protein deletion mutants[36]. Conjugation assays were performed in four replicates either in Miller's Lysogeny Broth (LB) or on LB agar plates (Fig. 1a, b) using the Rotor HDA from Singer instruments (Supp. Fig. 1). Our conjugation assay relies on the high-throughput replication capacity of the Rotor HDA to perform serial dilution arrays for every mutant tested (Supp. Fig. 2a–c). We aimed for a ratio of 1 donor for 1 recipient (Supp. Fig. 2d), although there was no significant variation in the conjugation score when a 5-fold excess of donor or recipient strains was used (Supp. Fig. 2e). Transfer rates were then evaluated by quantifying recipient and transconjugant cells after growth on solid media supplemented with appropriate antibiotics (Supp. Fig. 1). The growth of recipient and transconjugant cells was generally well correlated between the four replicates (average Pearson correlation coefficients of 0.74 and 0.81, respectively). For each deletion mutant, a conjugation score was assigned by dividing the number of transconjugants by the recipient bacteria (Supplementary Data 1). We estimate that our high-throughput screening method provides a semi-quantitative way to score the mutants with transfer rates ranging from $10^{-1}$ to $<10^{-6}$ (Supp. Fig. 2f). Overall, the scores followed a normal distribution (Fig. 1c, d) where the average conjugation score for transfer assays in solid medium ($0.86 \pm 0.08$) was slightly higher compared to broth ($0.81 \pm 0.09$). These results are consistent with published transfer rates of wild-type TP114 under the same conditions[23,26]. A total of 18 recipient strains representing low, average, and high conjugation scores in broth and solid conditions were selected for independent validations using a standard manual conjugation protocol instead of the high-throughput conjugation assay. For assays performed with the high-throughput conjugation protocol, gene deletion mutants with lower conjugation scores (Supplementary Data 1) also tend to have a lower conjugation rate using a standard conjugation protocol compared to wild type recipient strain (Supp. Fig. 3). Gene deletions with a conjugation score around or greater than the average transferred similarly to wild type in the manual protocol (Fig. 1e, f and Supp. Fig. 3). Taken together, these results suggest that our high-throughput method reliably identified recipient cell genes that impair TP114 conjugative

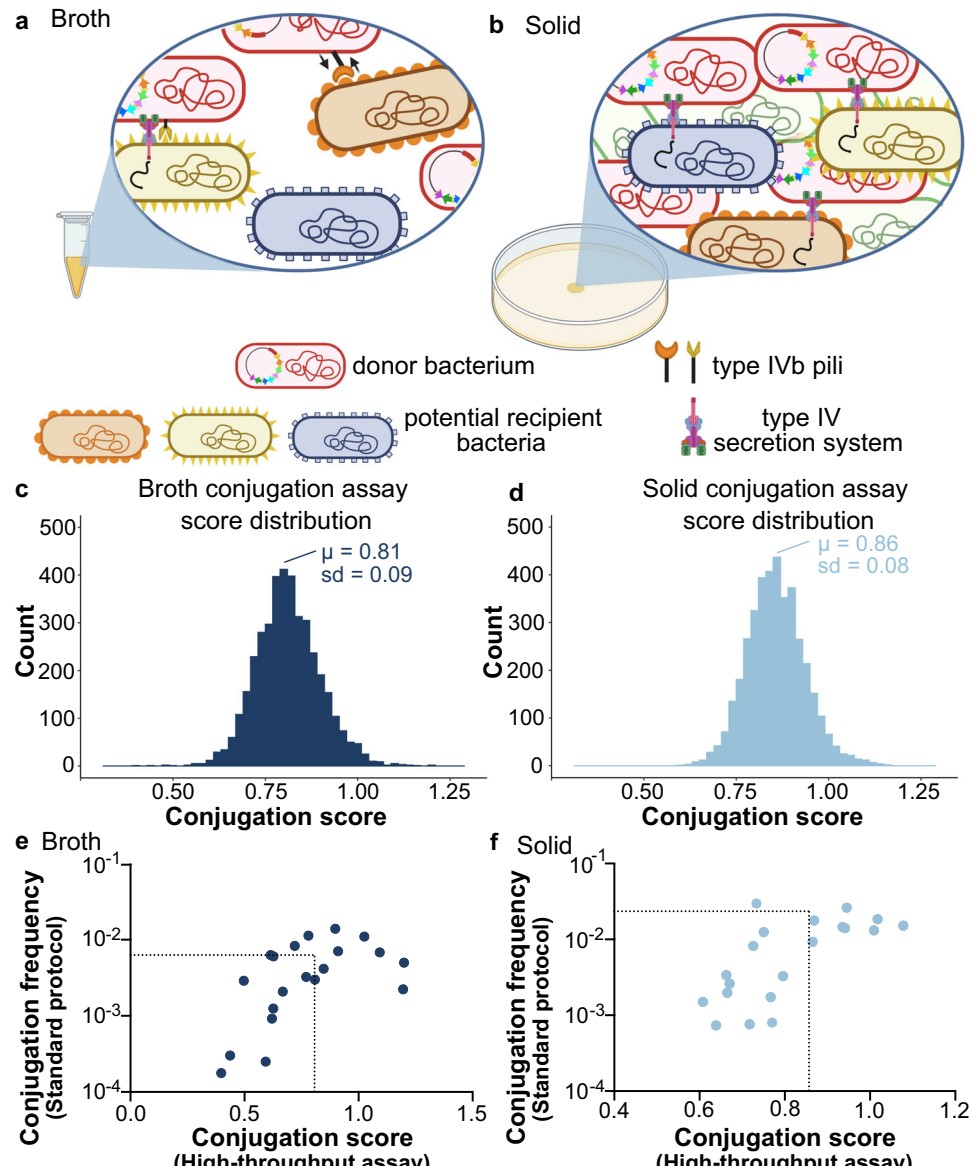

**Fig. 1 High-throughput conjugative transfer assay of TP114 in broth or on solid media. a** Bacterial conjugation occurring in broth requires Mating Pair Stabilization (MPS) provided by type IVb pili of TP114 to bring cells together and keep them in close contact during plasmid transfer. In this condition, the conjugative pilus and the accessory type IVb pilus will recognize specific molecules on the recipient's outer membrane. Created in Biorender.com. **b** Bacterial conjugation on solid media provides high cell density and proximity between donor and recipient cells enabling mating pair formation without the need for MPS via the type IVb pili. A high-throughput conjugation assay of TP114 into ~4000 *E. coli* single-gene deletion mutants yielded a normal distribution when mating was performed in broth (**c**) or on solid media (**d**) when plotting conjugation scores obtained from the ratio of the growth of transconjugants by recipient bacteria ($n = 4$). **e, f** Transfer rates of manually validated hits with low, medium, or high conjugation scores. The dotted lines represent the conjugation rates measured with a wild type recipient strain or the average conjugation score obtained in the high-throughput assay. The circles show individual values for each dataset ($n = 2$).

DNA transfer, but no gene that increases conjugation rates were confidently asserted.

**Conjugative transfer of TP114 towards LPS biosynthesis mutants is impaired in broth but not on a solid medium.** We next analyzed pathways affecting conjugation in our high-throughput assays in broth or on a solid medium. Instead of looking for specific genes, we used a global approach based on *Z* score statistics. Briefly, we converted the high-throughput conjugation scores into Z scores (Fig. 2a, b) and investigated only those below −1.96, which corresponds to *p* values ≤ 0.05. We chose to concentrate on elements found at the lower end of the

distribution, as we posited that they could reveal key receptors facilitating the conjugative transfer of TP114.

Z-scores represent the relative position of individual data points compared to the distribution of the dataset. Because the broth and solid mating distributions are similar, the number of hits is expected to be comparable for both screens, which represent 82 genes per condition in our case. From these, only 9 genes were shared between the two conditions, including *acrA*, *acrB*, and *tolC* of the AcrAB efflux pump as well as *carB*, *dsbA*, *glyS*, *pstC*, *recG*, and *sbcB* (Fig. 2c, d, Supplementary Data 1). We then performed gene ontology analysis to identify significantly enriched biological pathways and molecular functions associated with low conjugation scores (Fig. 2e). No gene ontology terms

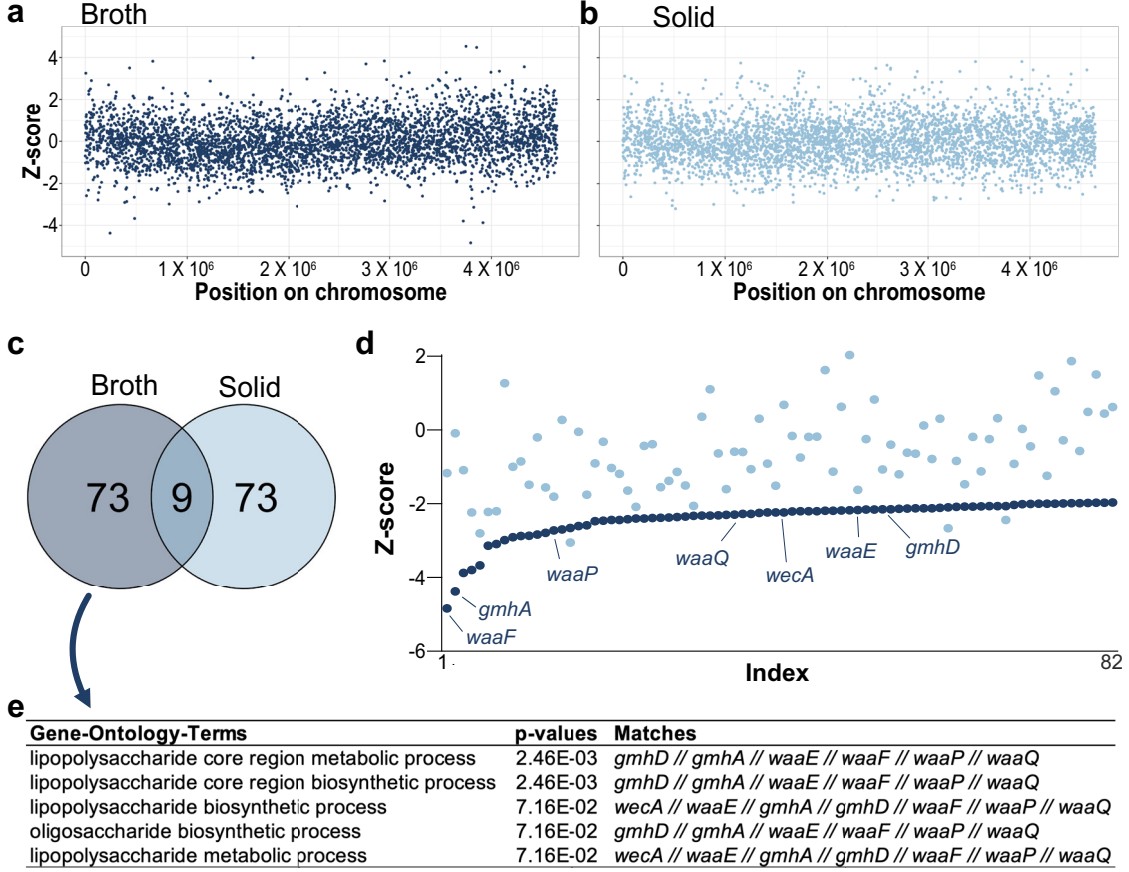

**Fig. 2 Genes implicated in the LPS biosynthesis pathway are specifically important for conjugation in broth. a**, **b** Conjugation scores for the mating in broth (**a**) and on solid media (**b**) were transformed into $Z$ scores and are represented according to the gene position on the chromosome. **c** Venn diagram representing the genes decreasing the efficiency of the conjugation ($Z$ scores < −1.96) in broth (dark blue) or on a solid support (light blue). **d** In this assay, 82 genes decreased the efficiency of the conjugation in broth ($Z$ scores < −1.96; $Z$ scores in broth (dark blue) and on solid (light blue) for these 82 genes are plotted according to the scores for the matings in broth **e**, gene ontology terms enriched for genes deletions affected when conjugation is performed in broth compared to a solid medium are associated with LPS biosynthesis.

were found for genes with low conjugation scores in solid matings when applying a Benjamini-Hochberg false discovery rate correction. In contrast, when the same analysis was performed for genes impaired during TP114 transfer in broth, LPS biosynthesis pathways were enriched (Fig. 2c–e), emphasizing their importance in this specific context.

**PilV variants recognize receptor structures in the LPS of *E. coli*.** Our previous work identified the T4Pb of TP114 as a major player in MPS in broth but not in solid medium[26]. This, combined with the gene ontology term enrichment analysis (Fig. 2e) highlighted the importance of LPS for efficient conjugation in broth. We surmise that the accessory pilus likely interacts with LPS molecules at the surface of recipient cells. To investigate the specificities of the T4Pb adhesins, we used eight derivatives of TP114 each displaying a single *pilV* gene expressed from its endogenous locus[26] instead of wild-type TP114. Indeed, the presence of a shufflon exchanging different DNA cassettes at the end of the *pilV* gene could lead to a heterogenous cell population with plasmids encoding up to eight different PilV variants (Fig. 3a, b) that presumably act as T4Pb adhesins with distinct specificities.

From the outer membrane to the extracellular space, the LPS structure consists of lipid A, core-oligosaccharide (core-OS), and O antigen[37] (Fig. 3c, see Supplementary discussion for more details). Although ~178 O antigen assortments were described for

*E. coli*, the core-OS exhibit a high degree of structural conservation accounting for only five distinct core structures designated K-12, R1, R2, R3, and R4[38]. We tested recipient strains representing the five core-OS prototypes in manual conjugation assays performed in broth. These results showed that the eight adhesins display different specificities for structures present in one or multiple strains as detected by the formation of transconjugants (Fig. 3d). Furthermore, all prototype strains lacked a complete O antigen but yet were still able to produce transconjugants, suggesting that this structure is not recognized by PilV adhesins. In contrast, conjugation was abolished with ClearColi, an *E. coli* strain with a truncated LPS displaying only the lipid IVA[39] (Figs. 3c, d and 4i), implying that the receptors used by the T4Pb of TP114 are likely part of the core-OS of the *E. coli* LPS.

**Elucidation of receptor structure(s) for PilV variants using knockout mutants.** To identify the specific receptor structures recognized by different PilV variants, we performed manual conjugation assays using Keio collection mutants displaying LPS structures truncated at various positions (Fig. 4 and Supp. Fig. 4b, c). Successful transfer assays in broth thus indicate the presence of the structure recognized by the tested variant, while the absence of transconjugants implies that the receptor was lost (Figs. 4 and 5a). For example, *E. coli* BW25113 is phenotypically rough and unable to synthesize long O16 antigen because of an

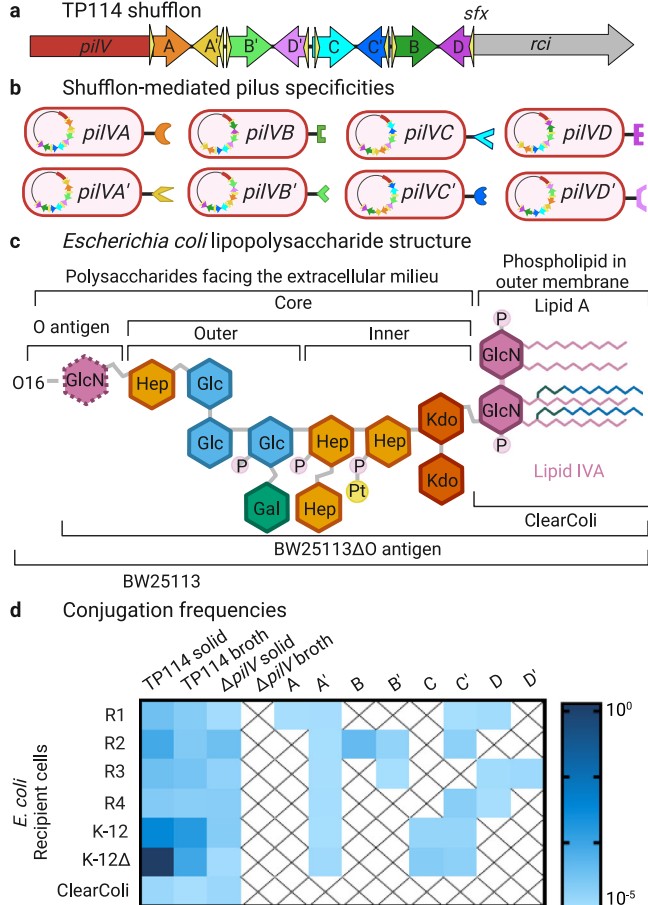

**Fig. 3 The LPS structures recognized by PilV adhesins are part of the core-oligosaccharide in *E. coli*. a** Schematic representation of TP114 shufflon where the 3'-end of *pilV* can undergo DNA rearrangement catalyzed by the shufflase (*rci*), a tyrosine recombinase that swaps the DNA segments comprised between two conserved recombination motifs (*sfx* sites). Only one possible conformation is shown. **b** DNA rearrangement of the shufflon in the donor bacterium allows the expression of eight different adhesins, presumably at the tip of the type IVb pilus. **c** General chemical structure of LPS from Gram-negative enterobacteria. LPS consists of the membrane-anchoring lipid A domain and a covalently linked polysaccharide or oligosaccharide portion. The depicted scheme represents the LPS architecture of *E. coli* BW25113 which corresponds to the distinct core structure termed K-12. Even though *E. coli* BW25113 lack long O antigen because the *wbbL* gene is interrupted by an IS5 insertion, we generate a mutant in which the *rfbABCD*, *wzxB*, *glf,* and *wbbHIJKL* genes were removed so that any of the O16 antigen molecules could be produced. The terminal glucosamine (GlcN) molecule is part of the O16 antigen but is still present in the BW25113ΔO antigen mutant since its incorporation only required the functions of *wecA* (*rfe*), an undecaprenyl-phosphate α-*N*-acetylglucosaminyl transferase, and *wzx*, an O antigen translocase, which are both in other operons elsewhere in the chromosome[64]. ClearColi only displays the lipid IVA which lacks the two secondary acyl chains and does not contain polysaccharides[39]. **d** Heat-map illustrating the transfer rates of wild-type TP114 along with the eight fixed *pilV* variants against various *E. coli* recipient cells. Transfer rates of the TP114Δ*pilV*::FLAG-*cat* mutant in which the variable region of *pilV* was replaced by a FLAG tag are also shown. All adhesins can transfer to at least one strain in broth. All conjugations were performed in biological triplicate with *E. coli* Nissle 1917Δ*dapA* as the donor strain. Cross marks indicate conjugation frequencies below the detection limit of the experiment ($1 \times 10^{-8}$). Created in Biorender.com.

IS5 insertion interrupting the rhamnosyl transferase WbbL coding gene[40]. Because of this, the *N*-acetylglucosamine added by WaaL constitutes the terminal sugar in the LPS of *E. coli* BW25113 (Figs. 3c and 4a, b). When the Δ*waaL* mutant is used as the recipient bacterium, the transfer of TP114 producing only the PilVC adhesin (TP114Δshufflon::*pilVC-cat*) is abolished. These results suggest that the receptor structure of PilVC is most likely the *N*-acetylglucosamine-β-(1-7)-heptose (Fig. 5b and Table 1).

Another example is the case of PilVA', for which the recognized structure appears to be highly conserved between the various core-OS prototypes as evidenced by the formation of transconjugants with all strains representing the five different *E. coli* core-OS types (Fig. 3d). In *E. coli* BW25113, all knockout mutants tested still produced transconjugants except for Δ*waaF*, Δ*waaP*, Δ*waaC*, Δ*waaE*, and Δ*gmhD* which are all involved in the formation of the core-OS (Fig. 4b–h, Supp. Fig. 4b, c and Fig. 5a). In addition, the phosphate group on the first heptose, added by WaaP, seems to be important for the recognition by PilVA' adhesin, but not the phosphate group on the second heptose, added by WaaY (Fig. 4c, d). It is also worth noting that the two heptose molecules containing an α-(1-3) linker and not an α-(1–2) linker are targeted since the Δ*waaQ* mutant does not contain the third heptose molecule but is still recognized by PilVA' adhesin (Fig. 4e). Finally, PilVA' no longer recognizes a Δ*waaF* mutant in which the second heptose molecule is absent (Fig. 4g). Taken together, these results suggest that the PilVA' receptor is most likely the L-*glycero*-D-*manno*-heptose-α-(1-3)-D-*glycero*-D-*manno*-heptose-4-phosphate (Fig. 5 and Table 1).

**Elucidation of receptor structure(s) for PilV variants using knock-in mutants.** The Keio knockout mutants can explain a limited number of receptor structures since *E. coli* BW25113 (K-12 LPS prototype) is recognized by only three PilV variants. However, all eight PilV variants can promote the transfer of TP114 to a strain exposing one of the five core-OS prototypes (Fig. 3d). To explore other potential receptor structures, each TP114 derivative was also tested with strains expressing one or two additional *waa* genes originating from other core-OS prototypes (Fig. 4j and Supp. Fig. 4e, f, h). These genes, responsible for the addition of a particular sugar in the LPS structure, were cloned in an arabinose inducible plasmid (pBAD30[41]) (Supplementary Table 1). As an example, when introducing pWaaID (bearing *waaI* and *waaD* genes from *E. coli* R3) in *E. coli* BW25113, this new strain expressing chimeric LPS produced transconjugants when mating in broth with the PilVD' adhesin, which was not the case with the wild-type strain. Assuming that WaaD is responsible for the incorporation of an *N*-acetylglucosamine via an α-(1-3) linker on a galactose molecule, we can infer that the *N*-acetylglucosamine-α-(1-3)-galactose is the receptor of PilVD' (Fig. 4j). Similarly, the receptor structure of PilVB was determined to be the *N*-acetylglucosamine-α-(1–2)-glucose (Supp. Fig. 4f, h). We also found that the galactose-β-(1–4)-glucose is a new receptor structure of PilVC' adhesin (Supp. Fig. 4e). The details surrounding the elucidation of all targets of adhesins can be found in the Supplemental Material and are summarized in Table 1.

**Discussion**

Conjugation plays a major role in bacterial evolution and antibiotic resistance. To achieve conjugative transfer, donor and recipient cells must establish close contacts leading to MPF[6,42,43]. Besides MPF, a generally overlooked step is Mating Pair Stabilization (MPS), which may be needed in unstable environments to

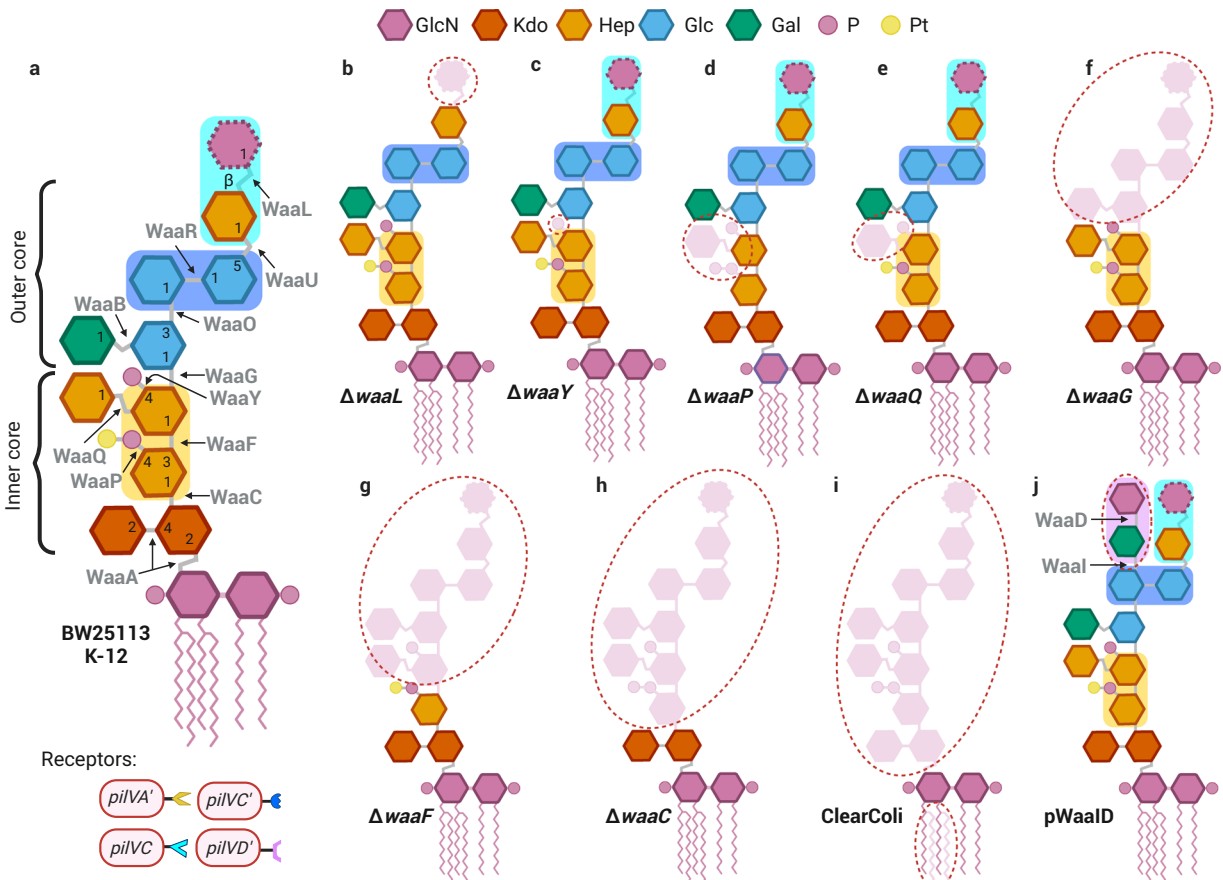

**Fig. 4 Comparison of LPS from *E. coli* strains and identification of adhesin receptors based on conjugation assays. a** The LPS structure of *E. coli* BW25113 with a K-12 core-OS is shown. The genes whose products catalyze the formation of each linkage are shown in gray. **b–i** The expected LPS structures of some knockout mutants[65] used in this study. The residues that are lost are depicted in pink and they are circled in red dotted lines. The expected LPS structure of *E. coli* BW25113 transformed with pWaaK (**j**). All glycoses are in the *a*-anomeric configuration unless stated otherwise. The specific receptor structures for the different PilV adhesins in the LPS molecules are indicated in colored rectangles. Abbreviations of monosaccharide residues: GlcN glucosamine, Kdo 2-keto-3-deoxyoctulosonic acid (3-deoxy-D-*manno*-octulosonic acid), Hep L-*glycero*-D-*manno*-heptose, Glc glucose, Gal galactose, P phosphate, Pt pyrophosphoethanolamine. Created in BioRender.com.

sustain the interaction between donor and recipient bacteria and allow successful DNA transfer[44]. MPS relies on adhesins displayed at the surface of the donor bacterium or on appendages such as pili that interact with specific structures on the outer membrane of recipient bacteria[26,45,46].

Conjugative plasmid TP114 has been highlighted for its remarkable transfer rate in the mouse gut microbiota[23]. Our group has recently demonstrated that a modified TP114 plasmid containing a CRISPR-cas9 module targeting *Citrobacter rodentium* could clear this pathogen from the mouse gut microbiota[24]. Interestingly, appendages such as an accessory type IVb pilus implicated in MPS are essential for the conjugation of TP114 in vivo or in broth but are dispensable for conjugation on a solid media[26]. Here, we investigated the recipient bacterium genes that impacted the transfer rate of TP114. To do this, we transferred TP114 into ~4000 *E. coli* deletion mutants using a high-throughput conjugation assay, both in broth and on a solid media (Fig. 1). Overall, our results show that few recipient cell genes significantly affected conjugation and no clear cellular function could be identified in the corresponding mutants except when MPS is important. Given the prominent role of the T4Pb for conjugation in unstable environments, genes that affected the transfer only in broth could inform us about the recipient cell receptors used by TP114. Our main finding is that LPS biogenesis in the recipient bacteria is crucial when conjugation is performed

in broth but is dispensable for mating on solid support (Fig. 2). Our screen, therefore, suggested that the adhesins of TP114 recognize structures within the LPS of the recipient bacteria.

The presence of the shufflon, which modifies the T4Pb adhesins used during conjugation[47], leads to the production of up to eight PilV variants[26], a population of cells harboring TP114 that will have different affinities for recipient cells (Fig. 3a, b). Therefore, to study the interaction between the PilV adhesins and the LPS structures at the surface of the recipient cells, we expressed the eight possible PilV variants individually. We observed that all eight PilV variants can promote the transfer of TP114 in at least one of the five core-OS prototypes found in *E. coli*, namely types R1, R2, R3, R4 and K-12 (Fig. 3d). Furthermore, by dissecting the core-OS region of the K-12 LPS using selected deletion mutants or by adding exogenous genes from other core prototypes, we found specific structures that are recognized by six of the eight PilV variants (Fig. 5 and Table 1).

Conjugative plasmid R64 also encodes a T4Pb with PilV adhesins that are conserved in TP114 (Supp. Fig. 5)[26]. Despite some point mutations changing between 5 to 36 amino acids in the variable region of PilV (Supp. Fig. 6), all homolog adhesins share the same receptor structures (Supp. Fig. 7). Until now, the specific ligand(s) of PilVA, PilVB', PilVC and PilVC' adhesins, encoded by conjugative plasmid R64, were known to be *N*-acetylglucosamine-β-(1-3)-glucose, *N*-acetylglucosamine-α-(1–2)-glucose,

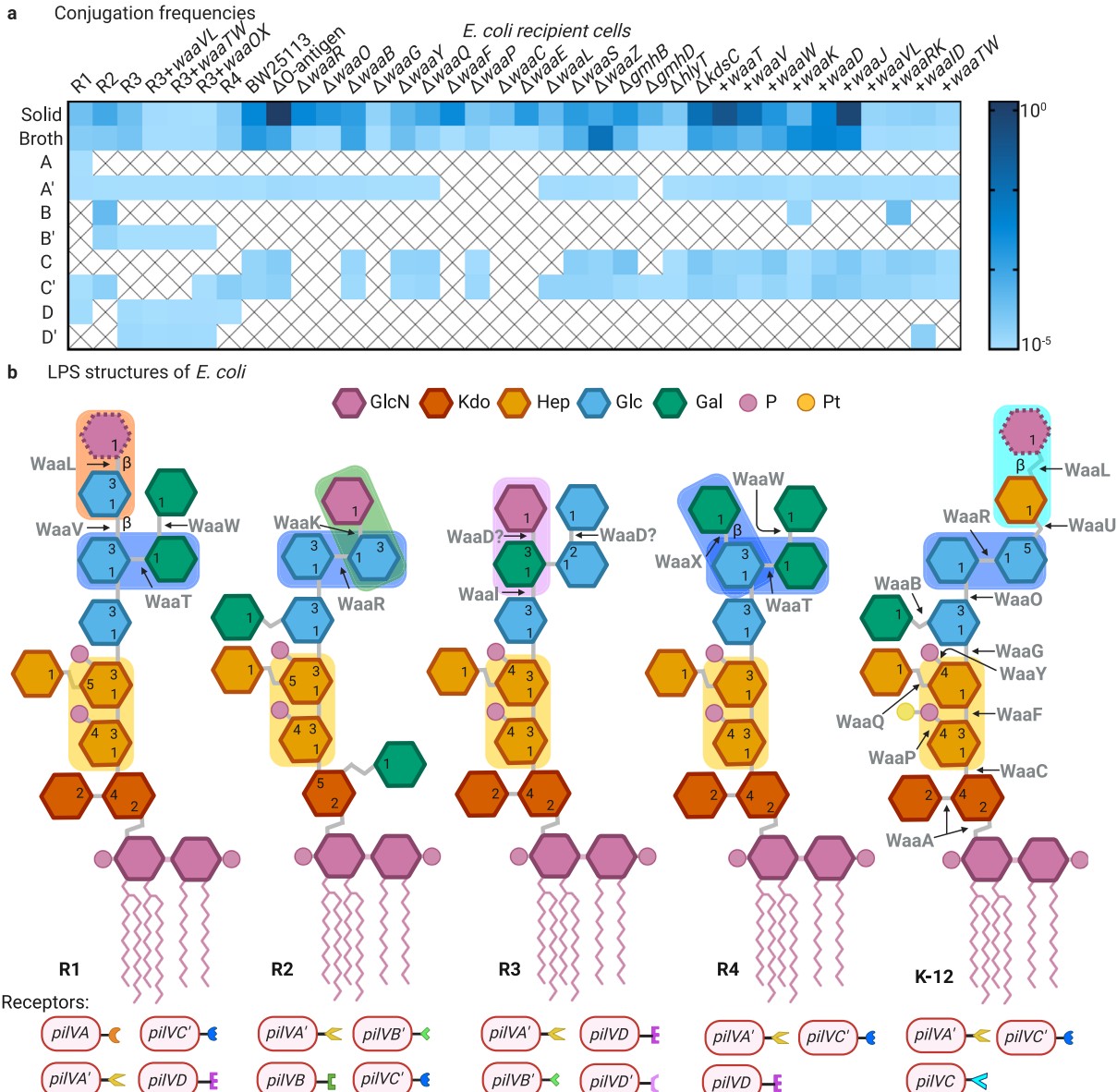

**Fig. 5 Comparison of enterobacterial LPS structures and identification of adhesin receptors based on conjugation assays. a** Heat-map illustrating the transfer rates of wild-type TP114 along with the eight fixed *pilV* variants against various *E. coli* recipient cells. All conjugations were performed in biological triplicate with *E. coli* Nissle 1917Δ*dapA* as the donor strain. Cross marks indicate conjugation frequencies below the detection limit of the experiment (1 × 10⁻⁸) in most cases. However, to avoid false-positive results, we considered that the conjugation results underneath 5.9 × 10⁻⁵ were not significant and were identified with a cross mark. **b** The LPS structure of *E. coli* R1, R2, R3, R4, and K-12 are shown. Details of the structures are taken from Heinrichs et al.[38], whereas numbers representing bond positions are taken from Bertani & Ruiz[66]. The genes whose products catalyze the formation of each linkage are shown in gray. All glycoses are in the *a*-anomeric configuration unless stated otherwise. The different variants of PilV allowing conjugative transfer in broth are depicted underneath each LPS structure. The specific receptor structures for the different PilV adhesins in the LPS molecules are indicated in colored rectangles. Abbreviations of monosaccharide residues: GlcN glucosamine, Kdo 2-keto-3-deoxyoctulosonic acid (3-deoxy-D-*manno*-octulosonic acid); Hep L-*glycero*-D-*manno*-heptose, Glc glucose, Gal galactose, P phosphate, Pt pyrophosphoethanolamine. Created in Biorender.com.

*N*-acetylglucosamine-β-(1-7)-heptose and glucose-α-(1–2)-glucose or glucose-α-(1–2)-galactose, respectively[30,31]. Our screen allowed us to infer the same receptor structures for PilVA, PilVC, and PilVC' adhesins. However, the *N*-acetylglucosamine-α-(1–2)-glucose was identified as the receptor structure of PilVB instead of PilVB' (see Supplemental material for details). Our conjugation results allow us to further infer the receptor structures of PilVA' and PilVD', as well as a new receptor structure of PilVC' (Fig. 5 and Table 1). Since one adhesin can have more than one receptor, such as PilVC' that can recognize three different targets, our conjugative screens are not sufficient to elucidate the receptor

structure of PilVB' and PilVD' yet. While we cannot exclude that other structures than the core-OS are recognized by these PilV variants, we do not expect them to target highly conserved structures or proteins since these adhesins allow conjugation only to specific strains. Further work will be needed to precisely identify their receptor(s).

When searching for other gene deletions negatively affecting the conjugation process, our genome-wide screen uncovered a potential role of the AcrAB efflux pump, as all three genes (*acrA*, *acrB*, and *tolC*) have a low conjugation score both in broth and on a solid medium (Supplementary Data 1, Figs. 1–2). One possible

**Table. 1 Summary of the structures recognized by PilV variants of TP114[a].**

| Adhesin | Receptor structure |
|---------|-------------------|
| PilVA | N-acetylglucosamine-β-(1-3)-glucose |
| PilVA′ | L-glycero-D-manno-heptose-α-(1-3)-D-glycero-D-manno-heptose-4-phosphate |
| PilVB | N-acetylglucosamine-α-(1-2)-glucose[†] |
| PilVB′ | unknown |
| PilVC | N-acetylglucosamine-β-(1-7)-heptose |
| PilVC′ | glucose-α-(1-2)-galactose |
|  | glucose-α-(1-2)-glucose |
|  | galactose-β-(1-4)-glucose |
| PilVD | unknown |
| PilVD′ | N-acetylglucosamine-α-(1-3)-galactose |

[a]Elucidation of each receptor structure is detailed in the Supplemental Material.
[†]Previously identified as the receptor structure of PilVB′ by (Ishiwa & Komano, 2003), but our results clearly demonstrate that it is the receptor structure of PilVB.

explanation is that the AcrAB pump is needed in the recipient cells to allow the expression of the newly acquired antibiotic resistance genes[48]. However, these hits were not validated using a manual conjugation protocol (Supp. Fig. 3). Our high-throughput assay differs from the standard protocol by the length of the conjugation step, with conjugation lasting 6 h and 2 h, respectively. During a 2 h conjugation assay, retransfer events (where transconjugants act as donor cells) are limited, while they are potentially more prevalent during a 6 h mating experiment required to obtain sufficient sensitivity and accuracy in the high-throughput screen. It is therefore possible that the AcrAB efflux pump affects the conjugation of TP114 in the donor cells. Incidentally, this suggests that there may be genetic determinants in the donor cells that are important for the transfer of TP114. A high-throughput experiment looking at the effect of gene deletions in donor cells has also highlighted the AcrAB efflux pump as a potential factor affecting the conjugative transfer of F-plasmids[1]. More work will be needed to identify donor cell genes affecting conjugation and understand the potential role of AcrAB-TolC in this process. Overall, our results in conjunction with the work of Alalam et al[1]. underscore the utility of high-throughput assays for advancing our understanding of the biology of conjugative plasmids.

Our high-throughput screening data did not uncover any function or pathways that significantly enhanced conjugation rates as gene ontology analyses for genes displaying high Z scores (i.e., high conjugation efficiency) showed no term or pathway enrichment. Although individual genes have conjugation scores higher than average, these effects were not validated using a manual conjugation assay, which could be due to false-positive hits given that the variability of our high-throughput screen is greater for high conjugation scores. For instance, the average standard deviation of the conjugation scores in broth for low and high hits are 0.09 and 0.14, respectively. Nonetheless, there are a few genes that had a high Z score in both broth and solid matings, such as *amiC*, *hflC*, *glgS*, or *yicN*. Other technical details such as the different mating times between the high-throughput screen and the manual validation could also be in cause (Fig. 1e, f and Supp. Fig. 3).

Our high-throughput assay was limited to the analysis of the non-essential gene deletions present in the Keio collection as well as small RNA and small protein deletion mutants. Since conjugative plasmids entering recipient cells use the host machinery for their replication and gene expression, these processes are expected to be essential for both the host and the conjugative plasmid. Bacterial cells also often possess defense mechanisms to

protect them against foreign genetic elements, such as CRISPR[10] or restriction-modification systems[11]. Therefore, the presence of such genes in the recipient cells would most likely alter the efficiency of the conjugation process. However, *E. coli* BW25113 is devoid of CRISPR and restriction-modification systems[49]. Altogether, our screening results suggest that, besides essential functions of the bacterial cell that we could not test, very few dispensable cell functions altered the transfer of TP114 into recipient cells, except for the receptors found at the cell surface when stabilization is required. Considering that the receptor structures of PilV adhesins are conserved within other bacterial genera such as *Salmonella*, *Shigella*, *Klebsiella*, *Citrobacter*, and *Vibrio*[50], which include pathogenic organisms implicated in healthcare-associated infections[51], the eight PilV variants in the shufflon of TP114 could contribute to the establishment of an extended transfer host range. It is well known that the dissemination of IncI2 plasmid occurs in *Enterobacterales*, such as *E. coli*, *Salmonella*, and *Klebsiella*[52] but can also be transferred to some *Pseudomonadales*[53]. This extended host range could be exploited to develop or improve new approaches to fight antibiotic resistance by conjugation-mediated effector delivery by live biotherapeutics[24]. The search for conjugation or MPS inhibitors could also contribute to the fight against multidrug resistance dissemination.

## Methods

**Bacterial strains, plasmids, and growth conditions**. Bacterial strains and plasmids used in this study are listed in Supplementary Table 1. Bacterial strains were grown at 37 °C in Miller's Lysogeny Broth (LB) medium in an orbital shaker/incubator set at 225 rpm or on LB agar plates in a static incubator. All strains were preserved at −80 °C in LB broth containing 25% (vol/vol) glycerol. Cells with thermosensitive plasmids (pSIM6, pE-FLP) were grown at 30 °C. When appropriate, antibiotics were used at the following concentrations: 100 μg/ml ampicillin (Ap), 34 μg/ml chloramphenicol (Cm), 50 μg/ml kanamycin (Kn), 100 μg/ml spectinomycin (Sp), 50 μg/ml streptomycin (Sm), 4 μg/ml nalidixic acid (Nx). Diaminopimelic acid (DAP) auxotrophy was complemented by adding DAP at a final concentration of 57 μg/ml in the medium. To induce expression from the $P_{BAD}$ promoter, LB medium was supplemented with 1.0% L-arabinose.

**DNA extractions, PCR amplifications, and sequencing**. Plasmid DNA was prepared using the EZ-10 Spin Column Plasmid DNA Miniprep kit (Bio Basic) according to the manufacturer's instructions. Restriction enzymes were purchased from New England Biolabs. A detailed list of oligonucleotides used in this study can be found in Supplementary Table 2. PCR amplifications were performed using TransStart® FastPfu Fly DNA Polymerase (Civic Bioscience) or TaqB (Enzymatics) for DNA amplification and PCR screening, respectively. PCR products were purified using MagicPure DNA Size Selection Beads (Civic Bioscience) or Monarch PCR & DNA Cleanup Kit (New England Biolabs) following the manufacturer's recommendations prior to assembly or recombineering, respectively. Sequences of interest were confirmed by Sanger sequencing at the Plateforme de séquençage et de génotypage du Centre de Recherche du CHUL (Université Laval, QC, Canada).

**Plasmids and strains construction**. Plasmids containing *waa* gene(s) from different sources were constructed using the pBAD30 plasmid as a backbone to express genes from the $P_{BAD}$ promoter. Plasmids were assembled from purified PCR products using the 2X NEBuilder Hifi DNA Assembly Master Mix (New England Biolabs) according to the manufacturer's protocol.

Following assembly, constructs were treated for 30 min at 37 °C with DpnI to eliminate residual DNA templates before transformation into chemically competent *E. coli* EC100D*pir*+ as described in Green and Rogers 2013[54]. Plasmids were then introduced into bacterial strains of interest by electroporation using a Bio-Rad GenePulser Xcell apparatus set at 25 μF, 200 Ω, and 1.8 kV with 1-mm gap electroporation cuvettes. The BW25113ΔO antigen::Kn$^R$ strain was generated by recombineering using the pSIM6 plasmid and 40-bp homology regions as described previously[23,55,56]. When required, the introduced antibiotic resistance cassette was removed from the resulting construction by Flp-catalyzed excision using pE-Flp. All modifications were verified by PCR and Sanger sequencing.

**High-throughput conjugation assays**. The high-throughput conjugation assays used *E. coli* MG1655 harboring TP114ΔKn::Cm as the donor strain. The mating experiment was performed with the ~4000 single-gene deletion mutants from the Keio collection[15] and 141 small RNA/small protein deletions[36]. This procedure was executed with four biological replicates. The donor bacteria were grown for 18 h in LB broth, while the recipient strains were arrayed on LB agar plates at a density of 384 mutants per plate (12 plates in total). For conjugations in broth, 50 ul of the donor strain diluted to an optical density of 0.1 was added in 384-well plates, and the recipient strains were subsequently added from solid source plates using the Singer Rotor HDA (Singer instruments, United Kingdom). For solid conjugations, a solution of donor bacteria diluted at an optical density of 0.1 was co-spotted with recipient strains from solid source plates on antibiotic-free LB agar plates at a density of 384 colonies per plate. We aimed to obtain a ratio of 1 donor for 1 recipient for conjugation. After 6 h of conjugation at 37 °C, each conjugation plate was replicated using the Rotor HDA on LB agar selective plates containing the appropriate antibiotics to select transconjugants (Kn, Cm) and all recipients (Kn), including those that became transconjugants. In addition, each conjugation was sequentially replicated 16 times to create dilutions and assess the transfer rates. Each plate was imaged with the Phenobooth (Singer instruments, United Kingdom) and the growth density for each mutant was determined using a method described by French et al.[57]. Briefly, plates were divided into 384 squared-shape region-of-interests (roi) and the density within each roi was measured using Fiji[58]. Density values thus correspond to the total density of the dilution series for each Keio mutant. Conjugation scores for each mutant were calculated by dividing the density of the transconjugant plate by the density of the recipient plate. Conjugation scores were then converted to Z scores, Z score = (Conjugation score–screen mean)/screen sd. We then used a global approach to analyze our screening results. Based on the statistical distribution obtained from the Z scores, we selected the upper and lower 2.5% of genes that were the most distant from the dataset average (*p* values ≤ 0.05). Then, we extracted gene ontology terms for each gene and performed gene ontology term enrichment (biological process) with a Benjamini–Hochberg correction using *pathway tools* in Ecocyc[59,60].

**Standard manual conjugation assays**. The standard manual conjugation assays used either *E. coli* MG1655Nx$^R$ or *E. coli* Nissle 1917Δ*dapA* as the donor strain for the validation of the high-throughput assays or further analysis of the impact of LPS biogenesis. The use of either strain as the donor in conjugation resulted in similar transfer efficiencies of TP114 (Supp. Fig. 8). Solid and broth conjugations were done at 37 °C for 2 hours as described previously[26]. Mating partners were serially diluted in 1X Phosphate Buffered Saline (PBS) and spotted in triplicates on

LB agar selective plates containing the appropriate antibiotics to discriminate between donors (Sp, DAP, Kn, and Cm), recipients (no antibiotic) or transconjugants (Kn, Cm). Colonies were counted, and conjugation frequencies were calculated according to the number of transconjugants per recipient colony forming units (CFU). All tested *E. coli* recipient strains are listed in Supplementary Table 1. For experiments with strains containing an arabinose inducible plasmid, 1.0% L-arabinose was added to the cultures after 16 hours of growth and was maintained during the conjugation assays. All experiments were performed in biological triplicate using three independently grown cultures. For the heat-map representations, transfer rates identified with a cross mark indicate that results were underneath the detection limit of the experiment ($1 \times 10^{-8}$) and no transconjugant was obtained. However, to avoid false-positive results, we considered that the conjugation results underneath $5.9 \times 10^{-5}$ were not significant and were identified with a cross mark as well.

**Phylogenetic analysis**. The NCBI database was used to retrieve the amino acid sequence of PilV variants of TP114 (MF521836.2) and R64 (AP005147.1). Only the variable parts of PilV adhesin protein sequences were aligned using the MUSCLE v3.8.31 algorithm[61] in the SeaView 5.0.4 phylogenetic analysis software package[62]. The maximum-likelihood phylogenetic tree was generated from the alignment file according to the PhyML method with an LG 4-rate class model[63]. Branch-support values were calculated by bootstrap analysis using 1000 replicates.

**Statistics and reproducibility**. The high-throughput screen was performed in $n = 4$ biologically independent samples whereas the manual validations were performed in duplicate. For the high-throughput screen, we calculated the Pearson correlation coefficient for each pair of replicates of both recipient and transconjugant densities, using the CORREL() function in Excel. We then averaged the Pearson coefficients for each dataset to evaluate the overall correlation between our replicates. All other conjugation assays were performed in triplicate. All data are represented as mean ± Standard Error of the Mean (SEM). The values were subjected to statistical analysis using GraphPad Prism 9.5.1.

**Reporting summary**. Further information on research design is available in the Nature Portfolio Reporting Summary linked to this article.

## Data availability

The source data behind the high-throughput screen is available in Supplementary Data 1. The source data behind all heat maps representing the conjugation frequencies is available in Supplementary Data 2. All new plasmids were deposited in Addgene and are publicly available: pWaaD (210448), pWaaID (210449), pWaaJ (210450), pWaaK (210451), pWaaRK (210452), pWaaT (210453), pWaaW (210454), pWaaTW-R1 (210455), pWaaTW-R4 (210456), pWaaV (210457), pWaaVL (210458), pWaaX (210459).

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

## Acknowledgements

This work was supported by the Canadian Institutes of Health Research [CIHR #159817]. S.R. and J.-P.C. hold a Chercheur boursier senior or junior 1 fellowship, respectively from the Fonds de recherche du Québec- Santé (FRQS). J.-P.C is also supported by the Natural Science and Engineering Research Council of Canada [NSERC#5014521]. This work was also supported by a NOVA- FRQNT-NSERC grant help by J.-P.C. and S.R. [FRQNT#314063-NSERC#571440]. N.A. and A.C. are respectively supported by a doctoral and master's scholarship from the Université de Sherbrooke. We thank Frédéric Grenier for assistance with plasmid sequencing. We thank Nicolas Allaire Tanguay for the design and construction of plasmid TP114ΔKn::Cm used in the high-throughput screen. We also thank Kevin Neil and Daphnée Lamarche for their insightful comments on the manuscript.

## Author contributions

A.C. performed the high-throughput screening with the Rotor HDA from Singer instruments, as well as manually conjugative assays to evaluate selected genes of interest. J.-P.C. wrote the normalization script for the high-throughput screening image processing and analyzed the data. J.P. constructed the pWaa plasmids containing only one *waa* gene and performed roughly a quarter of the in vitro conjugation assays under the supervision of N.A. N.A. design experiments, constructed all the other plasmids, performed most in vitro conjugation assays and analyzed the data. N.A. S.R., and J.-P.C. wrote the manuscript with input from all the authors.

## Competing interests

S.R. has a financial interest in TATUM bioscience. S.R. and N.A. are also co-authors of a patent application entitled 'Probiotic bacterial conjugative system and therapeutic uses thereof' (WO2020010452A1). The remaining authors declare no competing interests.
