## [Peer Review File · Communications Biology]

Reviewers' comments:

Reviewer #1 (Remarks to the Author):

This paper describes screening a deletion collection for mutants which are unable to act efficiently as recipients in a conjugation assay in liquid but not solid matings. They primarily find LPS mutants and then go on to characterize the genetic interaction between LPS variants and T4Pb (important for mating pair stabilization). This is a super interesting and important question but unfortunately the high throughput screen is not described sufficiently and was not validated enough. Because of the follow-up experiments, I can see some of the mutants were validated but I cannot trust the original data set.

First, I will address how the HTS screen was done and analyzed. (Methods line 353-372)

1. For the liquid matings, the donors and recipients were added differently. The donor was 50 ul at OD 0.1 and the recipient was pinned into this. What was the ratio of donor to recipient (at least on average)? Were the recipients in excess (as this could influence the results)?
2. For the solid matings, I'm not sure what the source of the cells was. Was it liquid or solid and in what format? Were there equal numbers of donors and recipients?
3. Were both the liquid and solid matings incubated for 6 hours at 37C? So, in both cases transconjugants have a long time to divide and also mate into remaining recipients?
4. The matings were pinned onto Kan to select for recipients, but that is not true. Colonies on Kan would include both recipients and transconjugants. So, any calculation of mating efficiency would be $\text{transconjugants}/(\text{transconjugants}+\text{recipients})$. This might be okay as long as recipients are in excess throughout the experiment but I am very unclear on this as stated above.
5. The matings were sequentially pinned 16 times to do 'dilutions' which is a great idea but there is no validation of how well this worked. What does the resulting dilution curve look like? Looking at the plates shown in Suppl Fig 2, it is difficult to interpret. The liquid mating dilutions clearly show single colonies for latter dilutions. Were those used in the calculation? This would create a big artifact. When you spot cells at high density, the cells are limited in growth partly by the neighboring cells within (and without) the spot, whereas a single colony would grow relatively without limitations. When the cell density is then calculated, it is the density of the entire spot that is calculated and since a single cell will grow much more than a dense spot, the curve would be very non-linear. The linear range of this method must be determined. Were the dilutions pinned in a 6144 format? Doesn't that create a problem of possible mating on the plate when the colonies after overnight incubation can touch each other in some cases? There is also information missing about how the conjugation score was calculated. There is no script included except the reference to French, et al where they did a slightly different setup and the sentence in line 370 ends abruptly as if it is missing text.
6. How many times were the dilutions/experiments repeated? I see 4 replicates in the excel file, but I don't know if those were biological or technical replicates.
7. In summary, regarding the high through-put screening, I think some of the data can clearly be trusted but it is not quantitative. For example, DwaaD shown in Suppl Fig 2 is clearly defective. If the authors do not want to do the work needed to more thoroughly validate the system, why not score the mutants qualitatively and then do the individual matings (already done)?
8. The manual verification differed in two major regards. A different strain donor was used and the time of conjugation was only 2 hours. Why?

Minor comments

9. Line 48, there are no references for 'DNA replication and gene expression'.
10. The sentence (lines 120-122) is not clearly written.
11. Line 130-131: You cannot say this as there is little/no measure of robustness: 'Our analysis focused on genes that decreased conjugation efficiency given the robustness of the high-throughput screen for low hits'
12. Line 144-146. It would be more accurate to delete: '...suggesting that the accessory pilus likely interacts with LPS molecules at the surface of recipient cells. To investigate...' and replace it with 'We suggest T4Pb plays a role and thus...'
13. The discussion needs to be modified in light of the comments above.

Reviewer #2 (Remarks to the Author):

This manuscript is an interesting piece of work in which the role of recipient bacteria in conjugation is analyzed. The experimental procedure is innovative, with large-scale analysis using the KEIO collection and a Rotor HAD device. The results in which the authors focus the work are the lipopolysaccharides, whose role in the conjugation is not extremely surprising, but it has never been analyzed in such detail. Additional genes observed in the analysis could also have an important role. Overall, I think this is a valuable and useful work, with interest to the scientific community working in microbiology. Notwithstanding, I do have some comments.

ABSTRACT

L21. 'conjugation is the most widespread mechanism of HGT' is a big statement. Conjugation is important and has been very well characterized, but its relative role compared to other HGT mechanisms has not been analyzed in nature (at least that I know). I would modify the sentence.
L27. 'only few genes in solid medium'. According to the results, the number of genes affected in solid and in broth were the same (n = 82), correct?
L31. Add type IVb 'accessory' pilus of TP114.

INTRODUCTION

L36. Same as comment in L21.
L42. 'Genetic contribution' does not sound correct. Better genetic functions?
L43. When a bacterium carries a plasmid, the plasmid-encoded genes are automatically 'host genes'. The goal is to analyze the chromosome-encoded genes. I would change 'host genes' to 'genetic background' or 'chromosome-encoded genes'.
L54-L61. Personal suggestion: I would start the paragraph with the last sentence (L58: TP114 was isolated from an E. coli strain ...). Firstly, the overall description of the plasmid; Secondly, the specific conjugative characteristics.
L54: 'identified as' to 'identified encoding'?
L60: Reference to a paper of IncI2 plasmids.
L63: (...) implicated in mating pair stabilization, 'rather than in DNA transfer'. (to be sure that the reader non-familiar with this system knows that the T4Pb is not a DNA channel). A reference to Figure 1A and 1B.
L64: The authors refer to the T4SS of these I-complex plasmids. Some authors additionally categorize these systems in phylogenetically groups (MPF-F, -I, -T, ...) (Guglielmini, 2014, <https://doi.org/10.1093/nar/gku194>). Since the purpose of the work is the characterization of this specific T4SS, it might be interesting to add the MPF group of the TP114 plasmid, since the related MPFs could share the same characteristics. If it is the same as R64, I guess it is the MPF-I.
L85: Add the protein names of the receptors for F-like plasmids '(OmpW, OmpK36, OmpF)' so that the reader can relate their functions with that of other genes observed in the present study.
L92: '(...), except for deletion mutants impacting LPS biosynthesis, (...)'. I do not completely agree with the sentence, since it implies that there were only genes related to LPS biosynthesis. Indeed, most of the genes were not related to LPS. I would change: '(...), with an important role of LPS biosynthesis in broth' (or similar).

RESULTS

L109. Which four replicates? No replicates are mentioned in the text so far.
L114. If the score is calculated as transconjugants (Kn, Cm) / recipients (Kn), are values >1 an artifact?
L114. The readers would greatly benefit if they could check which genes are shown with high and low rates for future studies. I would refer here to the Supplementary Table with the list of genes. Some comments on which are the genes with greater or lower conjugation rates could be interesting?
L116. How many different recipients were selected? Which thresholds were used to determine low, average and high conjugation scores?
L123-L125. The fact that the methodology changed from 2 hours to 6 hours of mating, make the reviewer wonder why. Why did the authors modify the mating duration? The reviewer can think of different factors that could alter the results: (1) In 6h, the authors give more time to

transconjugants to become new donors, thus increasing the chances of conjugation events. (2) This might be even more important than expected in cases of 'hyper-fertility' after conjugation, in which some plasmids have been shown to produce a 'hyper conjugative' state just after being transferred due to the time needed to repress the conjugative machinery in the new cell. (3) Recipients from the KEIO collection could show different growth rates due to the different single mutations, isn't it?. In 2 hours, the differences of fitness between cells is limited, but in 6 hours some differences might be observed, increasing in some cases the number of recipients in the mating, and reducing it in others. (4) Do the authors know if the aminoglycoside phosphotransferase can produce cheaters growing without the aph gene? In 6h, there could be more time for cheaters to grow. In summary, why the change from 6h to 2h? Did the authors try some examples with the same mating duration?

L126. I would remove 'severely'

L135. Are LPS biosynthesis involved in solid media?

L142. The reviewer understands that being the only GO enriched, the work focuses on that specific result. However, would not be also interesting to mention something about the remaining genes? Or those 9 genes shared between both solid and broth?

L152. Unclear if the eight PilV could be expressed within the population of from the same bacterium cell: (...) DNA cassettes leading to a heterogenous population with plasmids encoding each of the eight different PilV that presumably all act (...). (or similar)

L155. Supp Fig 3 is important and could be combined with Figure 3 in the main text.

L156-157. The authors seem to suggest that the fact that the O antigen being more variable than the core-OS, makes the latter more likely to be the target of the PilV. However, this is not explicit in the text.

L162-165. It seems that this hypothesis (core-OS being the receptor rather than the O antigen) was discarded in the L156-157. Could this experimental result fit more just after L157?

L170. 'Productive transfer' to 'Successful transfer', 'Efficient transfer', ...

L180. 'Taken as another example the case of PilVA, ...' or clarifying that you just focus on other example?

L198. Refer to Fig 3c.

L167-L212. Throughout these two sections, the reader needs to constantly change between figures 4 and 5a. These figures could be easily combined making it easier to read. Likewise, fig 5b would fit better in figure 3, where the different core-OSs are presented, and their conjugation analyzed.

DISCUSSION

L227. 'appendages such as 'accessory' pili that interact...'

L234. 'genetic elements' to 'genes'

L278. Remove 'Interestingly,'

L289. Mention few examples of interesting genes?

L310. An interesting point to discuss is the universality of these receptors, and its involvement in the host-range of the plasmid. Seen that these structures are present in other genera, it seems plausible that the host-range of these plasmids could cover these same genera. Do the authors find the plasmid TP114 (or related ones) in these other taxa? If not, there might be other factors involved in the host-range despite being able to form the MPS. If yes, are the PilV a more distant?

L296. Something I find interesting is that the authors do find the same number of genes with an importance in conjugation in solid. Are these genes related to LPS? Any clue on their functions and hypothesis? Why are some genes important in solid but not in broth?

L296. It could be interesting to know if these LPS biosynthesis genes are sometimes encoded in plasmids, since it might imply complex interplays between different plasmids (e.g. making the host bad recipient to similar plasmids and avoid incompatibles, ...).

METHODS

L330. 'Molecular biology' seems so broad that even the next section 'plasmid and strain construction' could fit in. Change title to more specific one, e.g. 'DNA extraction, amplification and sequencing' (or similar)

L363. 'Normalized' means actually 'diluted', isn't it?

L365. Donor number is known [50 ul of 0.1 OD (600nm?)], but we ignore number of recipients. Is the proportion aiming to a 1:1 ratio?

L374. What is 'density' in this context?

FIGURES

Figure 1. The reviewer would consider clarifying combining figures 1a, 1b and 3a together in a figure that could help the description of T4Cb, the sufflon, etc, in the introduction, when it is first mentioned.

Fig 4 and 5. Text in general is very small (letters a-j, genes knocked out, upper legend).

Reviewer #3 (Remarks to the Author):

In the submitted manuscript, Allard et al. characterize the IncI plasmid TP114's T4Pb shufflon specificity for recipient cell LPS variants. This was achieved first through a mechanized, high throughput conjugation screen. Second, direct confirmation of mating rates in T4Pb mutants, as a function of E. coli LPS variant, are reported. These major points, as well as the additional the supplementary figures, represent a useful step forward in the field of conjugation. Line-by-line feedback is present below. No additional experiments are recommended, just modification of text, figures, and references.

Line 37 – Citation 1 does not back the statement being made. I recommend citing an appropriate review on horizontal gene transfer, that mentions conjugation as a dominant mode of transfer. A good example would be reference 26 (PMID: 11212334) from the review you cited as ref 2.

Line 38 – Citation 2 mentions aspects of conjugative functions (Section 1-4) and their potential for genetic engineering (Section 5,6). A much better citation would be ref 7 from this paper (PMID: 10707066). However, an alternative review on, say, R plasmid-based adaptation could also be appropriate.

Line 47-49 – I agree with this sentence, however, you observe no hits in these pathways and even note that the method (Knockout collection screening) and the strain you use (BW25113) you are using is suboptimal for studying these pathways. Perhaps this should be removed from the introductory section, while leaving the previous sentence of MPF/MPS.

Line 61 – Ref 17 and 20 are good reviews for the referenced systems. Are the primary research articles in ref 18,19 most appropriate for representing the depth of research on the RP4 system? Could you find a relevant review article? For example PMID: 12833168 or PMID: 20600283 (Note that RP4 and RK2 are both some of the Birmingham IncP-1 plasmid isolates – they are remarkably similar).

Line 86 – It appears that the authors are drawing a distinction between recognition determinants of IncI and the example of F type plasmids. Do you think that is relevant to bring up the fact the F and F-like plasmids do respond to recipient cell LPS modifications (PMID: 7854127)?

Figure 1ef – Please edit the figure labels and/or legend to indicate that e. is Broth and f. is solid. This makes sense from the way the figure is structured, but ambiguity should be avoided.

Line 123-125 – It appears that the method, as presented here, is not less reliable at identifying conjugation enhancer mutations, but rather completely unable. Adjust this sentence to reflect this, or refute.

Line 136/Figure 2e – What are your methods for GO analysis? These should be included in the methods section as the tool used and gene ontology group chosen (biological process, molecular functions, or cellular components) will determine the outcome.

Line 162-165 – Why is the clearcoli data relegated to the supplements, rather than the main figures?

Line 283-294 – Could this discrepancy seen with acrAB-toIC mutants be explained by the finding of PMID: 31123134? I'm inclined to believe that decreased conjugation scores in these mutants is the result of alterations in antibiotic selection behavior. Further, if you think that they impact donor ability, perhaps there should be overlap with the Farawell lab high throughput dataset (ref1).

Line 305 – Do you mean devoid of active CRISPR systems? This is an important distinction, because mutations that activate this (which exist), could be reflected in your experiments.

Line 373 – At the very least, provide an imageJ version number. Ideally, what functions were used?

Line 375 – "...using at least." Using at least what?

Point-by-point response to reviewers' comments:

We thank the reviewers for their time and consideration in evaluating our manuscript. The questions and comments of each reviewer are reproduced below in black while our point-by-point response can be found in red.

Reviewers' comments:

Reviewer #1 (Remarks to the Author):

This paper describes screening a deletion collection for mutants which are unable to act efficiently as recipients in a conjugation assay in liquid but not solid matings. They primarily find LPS mutants and then go on to characterize the genetic interaction between LPS variants and T4Pb (important for mating pair stabilization). This is a super interesting and important question but unfortunately the high throughput screen is not described sufficiently and was not validated enough. Because of the follow-up experiments, I can see some of the mutants were validated but I cannot trust the original data set.

First, we want to thank the reviewer for noting the importance of the question that we studied in this manuscript. We are currently writing a “Lab protocol” along with a detailed workflow (protocol.io) that we will submit for peer review through PLoS One. This separate publication will delve more deeply into methodological details that cannot all be covered in this work. However, the results of important controls are presented below, to better answer the questions and concerns of the reviewers.

First, I will address how the HTS screen was done and analyzed. (Methods line 353-372)
1. For the liquid matings, the donors and recipients were added differently. The donor was 50 μ L at OD 0.1 and the recipient was pinned into this. What was the ratio of donor to recipient (at least on average)? Were the recipients in excess (as this could influence the results)?

For all our experiments, we aimed for a 1:1 ratio between the donor and recipient cells. We evaluated this ratio by counting the number of CFUs corresponding to different OD_{600nm} values (1.0; 0.1 and 0.01) of the donor strain culture relative to the pinned recipient bacteria (see Rebuttal Figure 1). As shown below, the average ratio using an OD_{600nm} of 0.1 was $\sim 1.5 \pm 0.7$ donor cells per recipient cell.

Rebuttal Figure 1: Ratio of donor to recipient cells in high-throughput conjugation assays. Donor cells harboring conjugative plasmid TP114 were resuspended at an OD_{600nm} of 1.0, 0.1 or 0.01, and 50 μ L of these suspensions were added to a 384-well plate. A recipient strain (*E. coli* BW25113 Δ yejO – a deletion mutant of a pseudogene in *E. coli* K-12) was then pinned into the 384-well plates. CFUs were then counted directly from these wells (at T=0 of conjugation). Each bar shows the average ratio for 8 wells.

We also measured the “conjugation score” in control experiments where the donor-to-recipient ratio varied as expected from the growth rate differences between Keio collection mutants. To do this, we used three dilutions of donor cells harboring TP114 with a constant amount of the recipient strain *E. coli* BW25113 $\Delta yejO$. We did not see any significant variation in the conjugation scores when a 5-fold excess of donor or recipient strains was used compared to the desired ratio of 1:1 (Rebuttal Fig. 2).

Rebuttal Figure 2: Effect of donor-to-recipient ratio on conjugation efficiency. Donor cells harboring TP114 were resuspended at various OD_{600nm} and mixed with the *E. coli* BW25113 $\Delta yejO$ recipient strain to vary donor to recipient ratios. After a 6h conjugation time, transconjugants (Kn and Cm) and recipients (Kn) were selected and the transfer rate was measured as described in the manuscript.

Importantly, we also performed the experiments with a non-conjugative plasmid and neither transconjugants nor any recipient cells were obtained on the double selection (for instance, recipient cells that become tolerant to the antibiotic on the plasmid because of a high quantity of cells). This observation held true for both matings in liquid and on solid support.

2. For the solid matings, I'm not sure what the source of the cells was. Was it liquid or solid and in what format? Were there equal numbers of donors and recipients?

For the solid matings, the source of the donor cells was a liquid culture at an OD_{600nm} of 0.1. The liquid culture was pinned onto solid plates at a density of 384 colonies per plate. The recipient cells were next pinned onto the donor spots from a solid source as described above.

The source of cells for solid matings is now stated more clearly in the methods section and additional details about the high-throughput conjugation experiments are also provided (L425-431; L437-441).

3. Were both the liquid and solid matings incubated for 6-hour at 37°C? So, in both cases transconjugants have a long time to divide and also mate into remaining recipients?

This is a very good point. We first attempted a 2-hour mating time for the high-throughput assay, but we didn't recover enough cells to accurately measure transfer rates. Moreover, the dynamic range of this experiment did not allow us to confidently distinguish between “wild type” transfer rates and potential gene deletion mutants that affect conjugation efficiency. To address these challenges, we settled on a 6-hour mating time. This compromise allowed us to

obtain a sufficient detection limit while reducing the possibility of retransfer compared to longer mating times. However, it is possible that a portion of the transconjugants became donor cells during this time. This means that if the gene deletion found in the Keio collection mutant affects the strain's ability to transfer the conjugative plasmid (but not its ability to receive it), we could still see this deletion mutant as a hit in our screen. The manual validation would however eliminate these false positives as they were performed for 2-hour to address this issue and allow comparison with previously published data on TP114 (PMID: 32963323; PMID: 35293798). We identified mutants that were hits in the high-throughput screen but did not reconfirm manually (Supp. fig. 2). For example, all genes of the *acrAB* efflux pump (*acrA*, *acrB*, and *tolC*) were identified in both liquid and solid mating high-throughput screens but were not validated in the 2-hour manual assays. Interestingly, transfer rates are affected when the *E. coli* BW25113 Δ *acrA* strain is used as the TP114 donor (data not shown), and was also observed to increase conjugation in another publication (Reference 1 in the manuscript). This is now mentioned in the discussion (L314-316; L321-344). The impact of single gene deletions in the donor strain is the topic of another manuscript in preparation by our group.

4. The matings were pinned onto Kan to select for recipients, but that is not true. Colonies on Kan would include both recipients and transconjugants. So, any calculation of mating efficiency would be $\text{transconjugants}/(\text{transconjugants}+\text{recipients})$. This might be okay as long as recipients are in excess throughout the experiment but I am very unclear on this as stated above.

The selection for the recipient bacteria is indeed performed with kanamycin, which also allows the growth of the transconjugants along with the recipients. However, mating efficiency quantifies the conversion rate of recipients into transconjugants. To do so, transconjugants must be included in the numerator and denominator, as the kanamycin selection permits. For example, if 75 transconjugants were obtained out of 100 potential recipients, the conversion rate would be 75% [$75 \text{ transconjugants} / (75 \text{ transconjugants} + 25 \text{ unconverted recipients}) = 75\%$]. However, excluding the transconjugants from the denominator would not make sense [$75 \text{ transconjugants} / (25 \text{ unconverted recipients}) = 300\%$]. Another way to see this is that transconjugants are recipient cells that acquire the conjugative plasmid. We have modified the Methods section (L433-434) to prevent any misunderstanding.

5. The matings were sequentially pinned 16 times to do 'dilutions' which is a great idea but there is no validation of how well this worked. What does the resulting dilution curve look like? Looking at the plates shown in Suppl Fig 2, it is difficult to interpret. The liquid mating dilutions clearly show single colonies for later dilutions. Were those used in the calculation? This would create a big artifact. When you spot cells at high density, the cells are limited in growth partly by the neighboring cells within (and without) the spot, whereas a single colony would grow relatively without limitations. When the cell density is then calculated, it is the density of the entire spot that is calculated and since a single cell will grow much more than a dense spot, the curve would be very non-linear. The linear range of this method must be determined.

Our strategy involved sequentially pinning our conjugations 16 times to create a dilution effect, followed by measuring the pixel density of bacteria on the entire surface of the dilution series

for a given mutant strain. The rationale behind this approach was that the dilution effect would lead to differences in the total density across the entire series, depending on whether cells continued to grow throughout the dilution series or stopped growing at earlier stages. Additionally, measuring the density within 384 squares at the plate level is more accurate than attempting to measure the density of 6144 individual colonies.

To investigate the linear range of the method, we examined the dilution effect by quantifying the density of the 16 individual dilutions from a random set of wells in a plate (see Rebuttal Figure 3). Whether the series originated from 'high-density' or 'low-density' wells, the dilutions remained linear for at least the first 9 steps. In cases where the density reached a plateau at the 9th dilution, there was typically no growth or the growth was minimal.

Rebuttal Figure 3: Linear range of the serial dilution process. To determine the linear range of the serial dilution from our high-throughput assay, we randomly selected four high-density wells, four low-density wells and one well with colonies at later dilution spots from a recipient plate of the screen (liquid mating). We then measured the density of individual dilution spots in FIJI. (A) Density of the 16 individual dilution spots from the 9 wells. (B) Total density of the 9 wells. (C) Image of the selected wells.

As the reviewer suggests, later dilution spots will occasionally have colonies that could contribute to increasing the overall density of the selected square. We included an example (shown in black in Rebuttal Figure 3) that depicts unexpected growth in late dilutions, which we consider to be a clear artifact. Interestingly, the colonies in the later dilutions only have a modest impact, as evidenced by the fact that the black well remains similar to the low-density wells when considering the total density of the square. However, we can imagine rare cases where especially large colonies in the latter dilution could have a more substantial effect on total density. To minimize the potential impact of growth artifacts like this, we also performed the screening in four biological replicates.

Were the dilutions pinned in a 6144 format? Doesn't that create a problem of possible mating on the plate when the colonies after overnight incubation can touch each other in some cases?

Yes, the serial dilution process consists of 16 sequential pinning steps, starting from a 384-well plate and resulting in a final 6144-well format. After overnight incubation, there is a possibility that a large colony might come into contact with neighboring colonies that are also large enough. However, this would not significantly affect our density measurements for two reasons: 1) the time required to obtain large colonies implies that the incubation period is likely over or near completion, leaving no time for additional growth, and 2) cells in large colonies are typically in the stationary phase and their growth is more limited. Furthermore, mating between a large colony and pinned bacteria that haven't grown significantly is unlikely due to the greater distance separating them. The antibiotic used to select the transconjugants, whether bactericidal (killing them) or bacteriostatic (inhibiting their growth), would either prevent or create a substantial delay before any conjugation event leads to colony formation. In this scenario, the incubation period of the plate is likely to have already ended, with minimal impact on the results. If conjugation still occurs despite all this, our quantification approach measuring the density of the entire area of the dilution series would also help mitigate the problem (see black line in Rebuttal Fig. 3).

There is also information missing about how the conjugation score was calculated. There is no script included except the reference to French, et al where they did a slightly different setup and the sentence in line 370 ends abruptly as if it is missing text.

The sentence on line 370 (now L442) has been corrected, and additional details have been added to the method section to describe the analysis and the calculation of the conjugation scores. Although the setup in the study by French, *et al.* is slightly different, we used the same script here. Further details have been included in the method section (lines L437-441).

6. How many times were the dilutions/experiments repeated? I see 4 replicates in the excel file, but I don't know if those were biological or technical replicates.

The screen was performed in 4 biological replicates as now clarified in the text (L114 & L422).

7. In summary, regarding the high throughput screening, I think some of the data can clearly be trusted but it is not quantitative. For example, DwaaD shown in Suppl Fig 2 is clearly defective. If the authors do not want to do the work needed to more thoroughly validate the system, why not score the mutants qualitatively and then do the individual matings (already done)?

We estimate that our high-throughput screening method provides a semi-quantitative way to score the mutants. As shown in Rebuttal Figure 4, the high-throughput conjugation scores correlate with manually assessed transfer rates and present a working approximation of the conjugation efficiency. For this experiment, TP114 variants showing different transfer rates ranging from 10^{-1} to $<10^{-6}$ (Nicolas Allaire-Tanguay, personal communication) were used in the high-throughput screen and the manual protocol. The results suggest that conjugation scores track relatively well with manual transfer rates (Rebuttal Figure 4), with a detection limit of around 10^{-5} .

Nonetheless, we recognize that the high-throughput conjugation score has limitations, which we attempted to mitigate using different strategies:

- 1) The high-throughput assay was performed in four biological replicates to reduce the variability.
- 2) For each mutant, we used the density of the entire dilution series instead of quantifying individual colonies to reduce susceptibility to experimental noise (see Rebuttal Figure 3);
- 3) A “systems-level approach” was used to investigate the potential hits instead of focusing on individual hits. For instance, we looked at the entire hit list with GO-term enrichment to unveil the important processes for the transfer of TP114 into the recipient cells. Our results clearly pointed to LPS at the surface of the recipient cells as a driver of TP114 transfer in liquid media, which was confirmed further with follow-up experiments.

In sum, while the high-throughput screen is not fully quantitative, the conjugation scores were still helpful in identifying important biological processes for the transfer of a conjugative plasmid in a recipient bacterium. Analyzing large mutant libraries such as the Keio collection generally comes at the expense of optimal accuracy.

8. The manual verification differed in two major regards. A different strain donor was used and the time of conjugation was only 2 hours. Why?

Regarding the different donor strains:

E. coli MG1655 was used for the high-throughput screen while *E. coli* Nissle Δ*dapA* was used for the manual assays. There are different reasons for this change:

- 1) *E. coli* MG1655 was used for the high-throughput screen to have a genetic background similar to *E. coli* BW25113.
- 2) Previous work with TP114 used *E. coli* Nissle as the donor (PMID: 32963323; PMID: 35293798). We wanted to be consistent with previous studies and facilitate comparisons for the manual assays.

- 3) We wanted to use an auxotrophic $\Delta dapA$ strain for conjugations towards the *E. coli* R1, R2, R3, and R4 prototype strains since they don't have any selective markers. We already had this mutant available in *E. coli* Nissle.
- 4) We understand that the nature of the donor could result in a difference in the conjugation efficiency. However, as shown in Rebuttal Figure 5, the conjugation efficiency is very similar whether *E. coli* MG1655, BW25113 or Nissle independently of the donor and recipient combination.

For these reasons, we don't think that using two different donor strains impacts our conclusions.

Regarding conjugation time:

As mentioned previously (see the response to comment 3), a 6-hour conjugation time was a compromise to obtain sufficient sensitivity in the assay with the drawback that some transconjugants can in turn become donor cells and promote the retransfer of TP114. Manual validations were thus performed with a 2-hour mating time to validate the results.

Minor comments

9. Line 48, there are no references for 'DNA replication and gene expression'.

We now added references for both mechanisms (now L49-50).

10. The sentence (lines 120-122) is not clearly written.

We reworded the sentence to clarify our point (now L130-134).

11. Line 130-131: You cannot say this as there is little/no measure of robustness: 'Our analysis focused on genes that decreased conjugation efficiency given the robustness of the high-throughput screen for low hits'

We have removed this sentence now and have rewritten this paragraph to better highlight the more global approach to analysis that we used to look at the screening results (now L136-142).

12. Line 144-146. It would be more accurate to delete: ‘...suggesting that the accessory pilus likely interacts with LPS molecules at the surface of recipient cells. To investigate...’ and replace it with ‘We suggest T4Pb plays a role and thus...’

This modification has been made (now L172-176).

13. The discussion needs to be modified in light of the comments above.

The discussion has been modified according to the comments above (L321-322; L335-353).

Reviewer #2 (Remarks to the Author):

This manuscript is an interesting piece of work in which the role of recipient bacteria in conjugation is analyzed. The experimental procedure is innovative, with large-scale analysis using the KEIO collection and a Rotor HAD device. The results in which the authors focus the work are the lipopolysaccharides, whose role in the conjugation is not extremely surprising, but it has never been analyzed in such detail. Additional genes observed in the analysis could also have an important role. Overall, I think this is a valuable and useful work, with interest to the scientific community working in microbiology. Notwithstanding, I do have some comments.

ABSTRACT

1. L21. 'conjugation is the most widespread mechanism of HGT' is a big statement. Conjugation is important and has been very well characterized, but its relative role compared to other HGT mechanisms has not been analyzed in nature (at least that I know). I would modify the sentence.

We thank the reviewer for pointing this out. We corrected the sentence for 'Bacterial conjugation is a major horizontal gene transfer mechanism.' (L21).

2. L27. 'only few genes in solid medium'. According to the results, the number of genes affected in solid and in broth were the same (n = 82), correct?

The reviewer is right; the same number of genes impacted the transfer of TP114 in both liquid and solid environments. This is due to the Z-score statistical method that was used to analyze the data. Z-scores quantify how individual data points compared to the overall dataset. Because the datasets for liquid and solid environments display a normal distribution with similar characteristics (number of data points, along with similar mean value and standard deviation for the dataset), the number of hits below a specific threshold is therefore expected to be similar. However, it is important to note that the lowest Z-score for liquid interactions was -4.84 (*ΔwaaF*), compared to -3.2 (*ΔybbK*) for solid ones. Additionally, in solid environments, the genes that reduced transfer rates did not appear to have any specific cellular function. Here the difference is that genes lowering the transfer rate in solid were not associated with a particular cellular function. We now have changed the sentence to: 'observed that recipient genes impairing transfer rates when conjugation occurs on a solid medium were not associated to a specific cellular function.' (L27-30).

3. L31. Add type IVb 'accessory' pilus of TP114.

We made this change (now L33).

INTRODUCTION

4. L36. Same as comment in L21.

We changed the sentence to 'Conjugation is a promiscuous and major mechanism of horizontal gene transfer in bacteria.' (now L37)

5. L42. 'Genetic contribution' does not sound correct. Better genetic functions?

We changed 'contribution' to 'function' (now L42).

6. L43. When a bacterium carries a plasmid, the plasmid-encoded genes are automatically 'host genes'. The goal is to analyze the chromosome-encoded genes. I would change 'host genes' to 'genetic background' or 'chromosome-encoded genes'.

We changed 'host' to 'chromosome-encoded' (now L44).

7. L54-L61. Personal suggestion: I would start the paragraph with the last sentence (L58: TP114 was isolated from an E. coli strain ...). Firstly, the overall description of the plasmid; Secondly, the specific conjugative characteristics.

We reordered the ideas of this paragraph accordingly (now L55-57).

8. L54: 'identified as' to 'identified encoding'?

We added the word encoding in the sentence (now L58).

L60: Reference to a paper of IncI2 plasmids.

We added a reference about the IncI2 plasmids (now L56).

9. L63: (...) implicated in mating pair stabilization, 'rather than in DNA transfer'. (to be sure that the reader non-familiar with this system knows that the T4Pb is not a DNA channel). A reference to Figure 1A and 1B.

We added the reference to Fig. 1 and modified the sentence (now L66-70).

10. L64: The authors refer to the T4SS of these I-complex plasmids. Some authors additionally categorize these systems in phylogenetically groups (MPF-F, -I, -T, ...) (Guglielmini, 2014, PMID: **24623814**). Since the purpose of the work is the characterization of this specific T4SS, it might be interesting to add the MPF group of the TP114 plasmid, since the related MPFs could share the same characteristics. If it is the same as R64, I guess it is the MPF-I.

Surprisingly, the T4SS of TP114 is not part of the MPF-I but is rather part of the MPF-T since all the genes encoded are closely related to the ones of the pTi plasmid. We now included this information at the beginning of the paragraph (L66-67).

11. L85: Add the protein names of the receptors for F-like plasmids '(OmpW, OmpK36, OmpF)' so that the reader can relate their functions with that of other genes observed in the present study.

We now added the types of outer membrane protein recognized by F plasmid family members (now L96).

12. L92: '(...), except for deletion mutants impacting LPS biosynthesis, (...)'. I do not completely agree with the sentence, since it implies that there were only genes related to LPS biosynthesis. Indeed, most of the genes were not related to LPS. I would change: '(...), with an important role of LPS biosynthesis in broth' (or similar).

We changed the sentence accordingly (now L102-104).

RESULTS

13. L109. Which four replicates? No replicates are mentioned in the text so far.

We now specified that the mating experiments were performed 4 times at the beginning of the paragraph (L114) as well as in the Material and Methods section (L422).

14. L114. If the score is calculated as transconjugants (Kn, Cm) / recipients (Kn), are values >1 an artifact?

Values greater than 1 for the conjugation score are indeed likely to be a side effect of the detection method that can be explained by different factors such as technical or biological “noise” in the experiment or biological amplification. For example, transconjugants become donor cells and could perform a limited number of transfer events in the dilutions on the plate used to select the transconjugants, hence slightly increasing the conjugation score. In any case, the conjugation scores remain well correlated with the transfer rates measured manually (see Rebuttal Fig. 4) and our conclusions are not affected.

15. L114. The readers would greatly benefit if they could check which genes are shown with high and low rates for future studies. I would refer here to the Supplementary Table with the list of genes. Some comments on which are the genes with greater or lower conjugation rates could be interesting?

We added the reference to Supplementary Data 1 (now L123), so the readers can consult the conjugation scores associated with each gene deletion mutant. We used a global approach to look at the hits from the screen, as group of genes gives a more comprehensive overview of the cell processes that impacted conjugation. Since LPS biogenesis was the only clear biological process coming out of this global approach, we decided to immediately follow up with LPS mutants instead of speculating on individual genes.

That being said, another group of genes (*acrA*, *acrB*, *tolC*) which are hits in both liquid and solid matings are also discussed (L311-344).

16. L116. How many different recipients were selected? Which thresholds were used to determine low, average and high conjugation scores?

We selected 18 different recipient strains both in broth and solid condition for the manual validations. Low and high hits were selected from the hit lists, while genes displaying average conjugation scores were chosen randomly in the rest of the dataset. This information has been added to the text (L127-128).

17. L123-L125. The fact that the methodology changed from 2 hours to 6 hours of mating, make the reviewer wonder why. Why did the authors modify the mating duration? The reviewer can think of different factors that could alter the results: (1) In 6h, the authors give more time to transconjugants to become new donors, thus increasing the chances of conjugation events.

(2) This might be even more important than expected in cases of ‘hyper-fertility’ after conjugation, in which some plasmids have been shown to produce a ‘hyper conjugative’ state just after being transferred due to the time needed to repress the conjugative machinery in the new cell.

(3) Recipients from the KEIO collection could show different growth rates due to the different single mutations, isn’t it?. In 2 hours, the differences of fitness between cells is limited, but in 6 hours some differences might be observed, increasing in some cases the number of recipients in the mating, and reducing it in others.

(4) Do the authors know if the aminoglycoside phosphotransferase can produce cheaters growing without the *aph* gene? In 6h, there could be more time for cheaters to grow. In summary, why the change from 6h to 2h? Did the authors try some examples with the same mating duration?

(1-2) The possibility of retransfer was also raised by reviewer 1 (see response to comments 3 and 8). We initially aimed to perform the high-throughput screen using a 2-hour conjugation time. However, from 2-hour of mating, we did not recover enough cells for the assay to be sufficiently sensitive and reliable, which resulted in a very small dynamic range. We finally decided to use a 6-hour conjugation time for the high-throughput assay, knowing that retransfer was a possibility. To make sure that we were looking at the effect of the deletions on the recipient cell, we then performed the validations and follow-up experiments with a 2-hour conjugation time as performed in previous studies.

(3) As for the effect of the mutation on the growth of *E. coli*, our conjugation score does take this into account by dividing by the total amount of kanamycin-resistant cells (all potential recipient cells in the mating mixture). Therefore, if the density of transconjugants is low because the mutation impairs the growth of the cell, we would also measure a low density on the recipient plate (kanamycin-resistant cells) which would normalize the conjugation score.

(4) Finally, we have not seen any suppressor mutants arising without the *aph* gene. We also co-incubated cells harboring a plasmid with the *aph* gene but without an origin of transfer, but we did not see any growth on the double selection plates.

18. L126. I would remove ‘severely’

We removed the term severely (now L143).

19. L135. Are LPS biosynthesis involved in solid media?

In a previous study, (PMID: 35293798) we demonstrated that the absence of either the major (PilS) or the minor (PilV) pilins did not significantly impede the capacity of the plasmid to transfer between cells on a solid support. Since the biosynthesis of the T4P is not required on solid support, we would expect that the genes involved in LPS biosynthesis are not important in this condition. Indeed, none of the LPS biogenesis genes having a low conjugation score in broth are found in the list of low conjugation scores on solid support (Supplementary Data 1).

20. L142. The reviewer understands that being the only GO enriched, the work focuses on that specific result. However, would not be also interesting to mention something about the remaining genes? Or those 9 genes shared between both solid and broth?

As we have decided to take a more global approach for the analysis of the screen, we refrained from talking about specific genes. However, hits in both broth and solid conjugations are indeed interesting. We have now added a mention of those genes in the results (now L161-162).

21. L152. Unclear if the eight PilV could be expressed within the population of from the same bacterium cell: (...) DNA cassettes leading to a heterogenous population with plasmids encoding each of the eight different PilV that presumably all act (...). (or similar)

The sentence was modified as proposed by the reviewer (now L174-181).

22. L155. Supp Fig 3 is important and could be combined with Figure 3 in the main text.

We now combined Fig. 3 with the Supp Fig. 3 in the main text.

23. L156-157. The authors seem to suggest that the fact that the O antigen being more variable than the core-OS, makes the latter more likely to be the target of the PilV. However, this is not explicit in the text.

This is not exactly what we wanted to say. Since 1) all PilV variants were able to transfer to at least one of the prototype strains and 2) all prototype strains are lacking O-antigen then the structure recognized by the PilV variant is in all probability present in the core-oligosaccharide. This being said, some disaccharides present in the core-oligosaccharide can also be found in the O-antigen of some strains. We reworded the sentence to clarify our point (now L190-195).

24. L162-165. It seems that this hypothesis (core-OS being the receptor rather than the O antigen) was discarded in the L156-157. Could this experimental result fit more just after L157?

This information could be said after L157, but we think it fits better where it is because we describe how we analyzed our results to elucidate specific receptors of PilV adhesins.

25. L170. 'Productive transfer' to 'Successful transfer', 'Efficient transfer', ...

We changed the word 'productive' to 'successful' in all text.

26. L180. 'Taken as another example the case of PilVA, ...' or clarifying that you just focus on other example?

We changed the sentence according to the proposition of the reviewer (now L213).

27. L198. Refer to Fig 3c.

We now refer to Fig 3c since we fusion Fig. 3 and Supp. Fig. 3 (now L208).

28. L167-L212. Throughout these two sections, the reader needs to constantly change between figures 4 and 5a. These figures could be easily combined making it easier to read. Likewise, fig 5b would fit better in figure 3, where the different core-OSs are presented, and their conjugation analyzed.

We understand that constantly changing between figures 4 and 5a to follow the analysis of the results can be challenging for the readers. However, both figures are already large, and they cannot be fused to fit on a single page without becoming very small. For this reason, we decided to keep Figures 4 and 5a separate. For Figures 5b and 3, it would not be optimal to fuse them because figure 3 shows the structure of the LPS and the first conjugation results with the different pilV adhesins without identifying a precise structure yet. Figure 5b is best found as the last figure of the article since it summarizes the entire analysis of the different structures recognized by the pilV adhesins not only in *E. coli* BW25113 but also for all 4 LPS prototype strains. For this reason, we prefer to keep the figures as they were presented initially.

DISCUSSION

29. L227. 'appendages such as 'accessory' pili that interact...'

We changed the sentence accordingly (now L267-268).

30. L234. 'genetic elements' to 'genes'

Again, we changed the sentence accordingly (now L270).

31. L273. Remove 'Interestingly,'

The word interestingly has been removed (now L316).

32. L289. Mention few examples of interesting genes?

We have now added a sentence that mentions a few genes that showed high conjugation scores in both broth and solid mating experiments (now L345)

33. L310. An interesting point to discuss is the universality of these receptors, and its involvement in the host-range of the plasmid. Seen that these structures are present in other genera, it seems plausible that the host-range of these plasmids could cover these same genera. Do the authors find the plasmid TP114 (or related ones) in these other taxa? If not, there might be other factors involved in the host-range despite being able to form the MPS. If yes, are the PilV a more distant?

We now specify that IncI2 plasmids are found in *E. coli*, *Salmonella* and *Klebsiella* strains, but also in some *Pseudomonadales* (results that will be published in another article about the host range of TP114) (now L370-372). Many plasmids found in the PubMed database also display PilV genes identical or near identical to the one found in TP114.

34. L296. Something I find interesting is that the authors do find the same number of genes with an importance in conjugation in solid. Are these genes related to LPS? Any clue on their functions and hypothesis? Why are some genes important in solid but not in broth?

As stated in comment 2, the fact that we find a similar number of genes affecting conjugation on broth and on solid is expected because of the Z-score statistics used to analyze the datasets. Because the two datasets are similar with a comparable distribution, an identical or very close number of hits should be found (82) outside of -1.96 standard deviation from the mean of the population. Therefore, we looked at the 82 lowest conjugation scores for the broth and the solid matings and we performed a 'systems-level' analysis. We made some changes to the text to clarify our point (now L152-162; L335-339).

None of the genes negatively impacting the transfer rate of TP114 on solid media are implicated in LPS biosynthesis. We have started to investigate genes that are required in both liquid and solid conditions or on solid medium only. We think that it is still too early to speculate on their roles as this will be the topic of future work.

35. L296. It could be interesting to know if these LPS biosynthesis genes are sometimes encoded in plasmids, since it might imply complex interplays between different plasmids (e.g. making the host bad recipient to similar plasmids and avoid incompatibles, ...).

It is known that the O54 antigen in *E. coli* is encoded on a plasmid (PMID: 12045108) and temperate bacteriophages sometimes encode genes responsible for the modification of the O-antigen (PMID: 31778182).

METHODS

36. L330. 'Molecular biology' seems so broad that even the next section 'plasmid and strain construction' could fit in. Change title to more specific one, e.g. 'DNA extraction, amplification and sequencing' (or similar)

This section is now 'DNA extractions, PCR amplifications and sequencing' to be more specific (now L390).

37. L379. 'Normalized' means actually 'diluted', isn't it?

We changed the word 'normalized' to 'diluted' (now L425).

38. L365. Donor number is known [50 ul of 0.1 OD (600nm?)], but we ignore number of recipients. Is the proportion aiming to a 1:1 ratio?

Yes, we aimed for a 1:1 ratio for the conjugation, we added this information in the text (now L430-431). Review 1 also raised a similar point; please see our response to comment 1 from reviewer 1 for a more detailed answer.

39. L383. What is 'density' in this context?

The analysis process converts the colonies from the dilution series into 'integrated densities'. The process is detailed in the French *et al.* reference cited in the method section. While the

setup is a bit different, the same analysis process is used here. Briefly, we measure the light passing through the colonies and thus the densities that are measured are correlated to the 'absorbance' of light by the colonies. More details about the plate analysis have now been added in the method section (now L437-441).

FIGURES

40. Figure 1. The reviewer would consider clarifying combining figures 1a, 1b and 3a together in a figure that could help the description of T4Cb, the sufflon, etc, in the introduction, when it is first mentioned.

We understand why the reviewer made this suggestion. However, these descriptions are needed at different points in the manuscript, and when we tried to combine them into a single figure, in our opinion, it complicated the narrative structure of the manuscript.

41. Fig 4 and 5. Text in general is very small (letters a-j, genes knocked out, upper legend).

The figures were first formatted for a different journal. We now have changed all figures according to Communication Biology's instruction to the authors.

Reviewer #3 (Remarks to the Author):

In the submitted manuscript, Allard et al. characterize the IncI plasmid TP114's T4Pb shufflon specificity for recipient cell LPS variants. This was achieved first through a mechanized, high throughput conjugation screen. Second, direct confirmation of mating rates in T4Pb mutants, as a function of *E. coli* LPS variant, are reported. These major points, as well as the additional the supplementary figures, represent a useful step forward in the field of conjugation. Line-by-line feedback is present below. No additional experiments are recommended, just modification of text, figures, and references.

1. Line 37 – Citation 1 does not back the statement being made. I recommend citing an appropriate review on horizontal gene transfer, that mentions conjugation as a dominant mode of transfer. A good example would be reference 26 (PMID: 11212334) from the review you cited as ref 2.

A similar comment was made by reviewer 2, so we changed the sentence to: 'Conjugation is a promiscuous and major mechanism of horizontal gene transfer in bacteria' (L37).

2. Line 38 – Citation 2 mentions aspects of conjugative functions (Section 1-4) and their potential for genetic engineering (Section 5,6). A much better citation would be ref 7 from this paper (PMID: 10707066). However, an alternative review on, say, R plasmid-based adaptation could also be appropriate.

We thank the reviewer for pointing this out, and we have changed the citation 2 accordingly.

3. Line 47-49 – I agree with this sentence, however, you observe no hits in these pathways and even note that the method (Knockout collection screening) and the strain you use (BW25113) you are using is suboptimal for studying these pathways. Perhaps this should be removed from the introductory section, while leaving the previous sentence of MPF/MPS.

The use of *E. coli* BW25113 and the Keio collection does not allow us to screen all the mechanisms potentially affecting bacterial conjugation. However, these pathways are already known to influence the process. We think it is important for readers to understand that these factors could also play a role in bacterial conjugation, to introduce the reasoning underlying the strategy that we adopted. We also cover these pathways in the discussion section (L354-366).

4. Line 61 – Ref 17 and 20 are good reviews for the referenced systems. Are the primary research articles in ref 18,19 most appropriate for representing the depth of research on the RP4 system? Could you find a relevant review article? For example PMID: 12833168 or PMID: 20600283 (Note that RP4 and RK2 are both some of the Birmingham IncP-1 plasmid isolates – they are remarkably similar).

We thank the reviewer for this information, we have removed ref 18 and added the 2 references suggested.

5. Line 86 – It appears that the authors are drawing a distinction between recognition determinants of IncI and the example of F type plasmids. Do you think that is relevant to bring

up the fact the F and F-like plasmids do respond to recipient cell LPS modifications (PMID: 7854127)?

Once again, we thank the reviewer for pointing this out, we added this information along with the reference suggested (now L96-97).

6. Figure 1ef – Please edit the figure labels and/or legend to indicate that e. is Broth and f. is solid. This makes sense from the way the figure is structured, but ambiguity should be avoided.

We added appropriate labels for panels e and f to avoid any ambiguity.

7. Line 123-125 – It appears that the method, as presented here, is not less reliable at identifying conjugation enhancer mutations, but rather completely unable. Adjust this sentence to reflect this, or refute.

We believe that our method would be able to detect genes that increase transfer rates, but in the case of TP114, we simply did not find any genes that do so. As shown in the response to comment 7 from reviewer 1 (see Rebuttal Figure 4), we are indeed able to discriminate between transfer rates from 10^{-1} to 10^{-5} . The conjugation scores from our assay are equivalent to a transfer rate of approximately 10^{-3} , suggesting that there is a window to observe genes that increase conjugation rates. However, as stated in our discussion, the assay seems more variable at higher scores. Thus, we have noted that it is less reliable to find enhancer mutants but this does not necessarily rule out the possibility of identifying such mutants for other conjugative plasmids.

8. Line 136/Figure 2e – What are your methods for GO analysis? These should be included in the methods section as the tool used and gene ontology group chosen (biological process, molecular functions, or cellular components) will determine the outcome.

Details have now been added in the method section (L444-449).

9. Line 162-165 – Why is the clearcoli data relegated to the supplements, rather than the main figures?

Reviewer 2 also suggested putting those results in the main article. These results are now included in Figure 3.

10. Line 283-294 – Could this discrepancy seen with *acrAB*-*tolC* mutants be explained by the finding of PMID: 31123134? I'm inclined to believe that decreased conjugation scores in these mutants is the result of alterations in antibiotic selection behavior. Further, if you think that they impact donor ability, perhaps there should be overlap with the Farawell lab high throughput dataset (ref1).

We thank the reviewer for this observation. Indeed, the *acrAB* efflux pump may affect the antibiotic susceptibility of the recipient cells. Interestingly, the *acrA* mutant that we tested in 2-hour manual conjugations did not show a decrease in conjugation efficiency. However, when we test a Δ *acrA* donor strain, the conjugation efficiency is now decreased (data not shown). *acrB* and *tolC* were also found as hits in the Farawell dataset, which was generated using an F-

plasmid. We have now updated our discussion about the *acrAB* efflux pump to reflect this (now L314-334).

11. Line 305 – Do you mean devoid of active CRISPR systems? This is an important distinction, because mutations that activate this (which exist), could be reflected in your experiments.

We meant completely devoid and not just inactive, so no mutations could activate it.

12. Line 373 – At the very least, provide an imageJ version number. Ideally, what functions were used?

A brief description of the plate analysis pipeline has been added to the method section and finally, we used the FIJI program instead of ImageJ (now L437-441).

13. Line 375 – “...using at least.” Using at least what?

This is an error in the initial manuscript that we have now corrected (now L442). We apologize and thank the reviewer for pointing this out.

Here are the changes that we made to the figures:

Figure 3 | The LPS structures recognized by PilV adhesins are part of the core-oligosaccharide in *E. coli*.

As proposed by Reviewers #2 and #3, we combined Fig. 3 and Supplementary Fig. 3 to include the results of ClearColi in the main text. To do so, we included the new panel c

comparing the LPS structure of *E. coli* BW25113, BW25113 Δ O antigen and ClearColi and we combined the two heat maps of conjugation frequencies in one.

No major changes have been made in other figures, except enlarging the text and optimizing the colour for readers with colour blindness.

REVIEWERS' COMMENTS:

Reviewer #1 (Remarks to the Author):

I want to thank the authors for making such a clear and extensive response to my previous comments. I am quite satisfied with the additional information they have added to the manuscript. This work shows a clear mechanistic role for T4Pb variation and interaction with diverse LPS on E. coli recipients. The methods used are clearly explained and validated and the results are clear and conclusions well founded. Further, they include a modification of methodology which could be generally useful.

I have only a few minor comments:

1. I would suggest that the type of plasmid used is in the title: ie, 'IncI2 plasmid conjugation' as we know different plasmids have different features.
2. There is a typo in Fig S2f (rebuttal fig 4): transfert
3. Depending on future work, it might be a good idea to complement a few of the key Keio mutants identified since Alalam, et al (reference 1) saw significant secondary mutations in their screen. However, I don't think this is required for publication of this paper because a) you have many mutants which corroborate each other and b) you changed the LPS by adding genes from other variants which support the results. It is simply a suggestion for future work.

Reviewer #2 (Remarks to the Author):

This manuscript is an interesting piece of work in which the role of recipient bacteria in conjugation is analyzed. The experimental procedure is innovative, with large-scale analysis using the KEIO collection and a Rotor HAD device. The results in which the authors focus the work are the lipopolysaccharides, whose role in the conjugation is not extremely surprising, but it has never been analyzed in such detail. Additional genes observed in the analysis could also have an important role.

Overall, I think this is a valuable and useful work, and after the changes provided by the authors I find no strong reasons to decline its publication.

Point-by-point response to reviewers' comments:

We thank the reviewers for their time and consideration in re-evaluating our manuscript. The questions and comments of each reviewer are reproduced below in black while our point-by-point response can be found in red.

Reviewers' comments:

Reviewer #1 (Remarks to the Author):

I want to thank the authors for making such a clear and extensive response to my previous comments. I am quite satisfied with the additional information they have added to the manuscript. This work shows a clear mechanistic role for T4Pb variation and interaction with diverse LPS on E. coli recipients. The methods used are clearly explained and validated and the results are clear and conclusions well founded. Further, they include a modification of methodology which could be generally useful.

I have only a few minor comments:

1. I would suggest that the type of plasmid used is in the title: ie, 'IncI2 plasmid conjugation' as we know different plasmids have different features.

We have changed the title accordingly, which now reads **“Systematic investigation of recipient cell genetic requirements reveals important surface receptors for conjugative transfer of IncI2 plasmids”**.

2. There is a typo in Fig S2f (rebuttal fig 4): transfert

We thank the reviewer for pointing this out. This typo have been removed in the revised version of the manuscript.

3. Depending on future work, it might be a good idea to complement a few of the key Keio mutants identified since Alalam, et al (reference 1) saw significant secondary mutations in their screen. However, I don't think this is required for publication of this paper because a) you have many mutants which corroborate each other and b) you changed the LPS by adding genes from other variants which support the results. It is simply a suggestion for future work.

We thank the reviewer for this suggestion. We agree that the complementation of key Keio collection mutants could strengthen our conclusions, but as mentioned by the reviewer, there are already many mutants that corroborate each other. We save this suggestion for future work.

Reviewer #2 (Remarks to the Author):

This manuscript is an interesting piece of work in which the role of recipient bacteria in conjugation is analyzed. The experimental procedure is innovative, with large-scale analysis using the KEIO collection and a Rotor HAD device. The results in which the authors focus the work are the lipopolysaccharides, whose role in the conjugation is not extremely surprising, but it has never been analyzed in such detail. Additional genes observed in the analysis could also have an important role.

Overall, I think this is a valuable and useful work, and after the changes provided by the authors I find no strong reasons to decline its publication.

We thank the reviewer for agreeing to review our revised manuscript and for the constructive comments throughout the process.